# Functional beverage development from traditional Thai polyherbal tonic: Antioxidant-rich microcapsules and comprehensive sub-chronic toxicity assessment

Thammarat Kaewmanee[1☯], Acharaporn Issuriya[2☯], Piyapong Choochana[3], Pinanong Na-Phatthalung[4], Surasak Limsuwan[5*], Sasitorn Chusri[6*]

**1** Department of Food Science and Nutrition, Faculty of Science and Technology, Prince of Songkla University, Muang, Pattani, Thailand, **2** Division of Health and Applied Sciences, Faculty of Science, Prince of Songkla University, Hat Yai, Songkhla, Thailand, **3** College of Oriental Medicine, Rangsit University, Pathumthani, Thailand, **4** Division of Hematology and Oncology, Icahn School of Medicine at Mount Sinai, New York, New York, United States of America, **5** Traditional Thai Medical Research and Innovation Center, Faculty of Traditional Thai Medicine, Prince of Songkla University, Hat Yai, Songkhla, Thailand, **6** Biomedical Technology Research Group for Vulnerable Populations and School of Health Science, Mae Fah Luang University, Muang, Chiang Rai, Thailand

☯ Thammarat Kaewmanee and Acharaporn Issuriya, as co-first authors, played significant roles in this work, contributing equally to its development and execution.
* sasitorn.chu@mfu.ac.th (SC); surasak.l@psu.ac.th (SL)

## Abstract

This study aimed to optimize microencapsulation conditions for Phy-Blica-O (PBO), a traditional Thai polyherbal tonic, and to assess the safety of its consumer-accepted herbal tea formulation, Phy-Blica-D (PBD). PBO decoction and its phenolic-rich extract were spray-dried at different inlet temperatures (140°C, 180°C) and maltodextrin-gum Arabic ratios. Encapsulation efficiency was highest at 140°C with maltodextrin alone, while antioxidant activity was greatest in microcapsules prepared at 180°C with a 6:4 maltodextrin to gum Arabic ratio, as shown by DPPH, ABTS, and FRAP assays. PBD demonstrated strong *in vitro* antioxidant activities and was subsequently assessed in a 90-day subchronic toxicity study in Wistar rats. No treatment-related mortality, clinical abnormalities, organ toxicity, or hematological or biochemical disruptions were seen at doses up to 300 mg/kg/day, indicating a NOAEL above this level. These findings suggest that optimized PBO microcapsules and the PBD tea blend are safe, antioxidant-rich functional ingredients with strong potential for commercialization in complementary and integrative medicine.

## 1. Introduction

Functional foods have gained increasing global interest due to their potential role in disease prevention, promoting overall health, and enhancing well-being. Complementary and integrative medicine has extensively employed traditional herbal

**Data availability statement:** All relevant data are within the manuscript and its Supporting information files.

**Funding:** This research was supported by the Prince of Songkla University (Pattani campus) (Grant No. SAT601363S) and was partially supported by Mae Fah Luang University (Grant No. 651C5003). Mae Fah Luang University covered the article processing fees charge.

**Competing interests:** The authors have declared that no competing interests exist.

**Abbreviations:** ABTS, 2,20-azino-bis-3-ethylbenzthiazoline-6-sulfonic acid; DMSO, dimethyl sulfoxide; DPPH, 1,1-diphenyl-2-picrylhydrazyl; FRAP, Ferric reducing antioxidant power assay; GLP, Good Laboratory Practice; HED, human equivalent dose; NOAEL, The oral no-observed-adverse-effect-level; OECD, The Organization for Economic Cooperation and Development; TFC, The total flavonoid content; TPC, Total phenolic content.

formulations, particularly polyherbal preparations, to enhance overall health and well-being, including traditional herbal beverages [1]. Herbal beverages are a key part of functional foods and are known for their health benefits, such as boosting energy, alleviating fatigue and stress, enhancing immune function, and offering protection against several metabolic syndromes and age-related diseases [2–4]. The documentation of traditional knowledge combined with the scientific validation of health claims for these beverages is becoming crucial for fostering innovation and gaining consumer acceptance in this market.

Our prior research on traditional Thai polyherbal infusions utilized as rejuvenating formulations has identified several promising combinations that exhibit beneficial biological activities related to health, including antioxidant properties and a preventive effect against hyperlipidemia [5–8]. Among these formulas, *Phyllanthus emblica*-based functional herbal tea (THP-R016 or Phy-Blica-O (PBO)) possessed notable antioxidant properties and did not exhibit cytotoxic effects on *Vero* cells at concentrations up to 100 µg/mL, supporting its safety for oral applications [9,10]. Numerous studies have confirmed the pharmacological properties of various herbal constituents of PBO *in vivo*, demonstrating their value in functional foods [9]. However, *Allium sativum* and *Tinospora crispa*, which are present in PBO, impart a strong bitter taste and unpleasant odor, limiting their application in the functional food industry. The utilization of microencapsulation in PBO not only enhances the sensory profile but also protects heat-sensitive bioactive compounds from degradation, thereby improving the functional properties and shelf-life of the formulation [11]. Spray-drying was chosen due to its scalability, reduced operational costs, brief processing duration, and appropriateness for heat-sensitive herbal extracts when compared to other methods [11]. Additionally, maltodextrin (MD) and gum Arabic (GA) are common wall materials in spray-drying botanical extracts because of their excellent film-forming, emulsifying, and protective qualities [12–14]. MD helps achieve high encapsulation efficiency and maintains low viscosity, whereas GA improves stability and retains antioxidants. Using them together has been demonstrated to increase encapsulation yields and functional performance in earlier research [12–14].

Phy-Blica-D (PBD), an herbal tea blend based on PBO, has been developed with remarkable consumer acceptability and demonstrates gastroprotective effects with an oral no-observed-adverse-effect level exceeding 300 mg/kg body weight/day in the 28-day repeated oral dose toxicity test [15,16]. The calculated human equivalent dose value is 48.39 mg/kg/day, or around 600 mL/kg body weight/day, with no target organs being affected. Nonetheless, regulatory authorities require comprehensive safety evaluations, including sub-chronic oral toxicity studies, to ensure the safety of innovative functional foods meant for regular human intake. The 90-day repeated-dose toxicity test plays a crucial role in detecting possible chronic toxic effects, establishing safe dosage levels (NOAEL), and verifying the absence of target organ toxicity and biochemical irregularities.

To facilitate the ongoing advancement of PBO and PBD as herbal-based functional beverages rich in antioxidants, we developed a novel approach by transforming PBO into a functional ingredient using optimized microencapsulation techniques.

The conditions for encapsulation, including inlet temperature and wall material composition, were methodically adjusted to improve physicochemical stability and antioxidant effectiveness. This process could enhance the antioxidant properties of PBO and potentially increase consumer acceptance. Additionally, a consumer-acceptable herbal tea blend, PBD, was developed and tested for its antioxidant potential through a series of *in vitro* assays. Notably, a comprehensive 90-day sub-chronic oral toxicity study in rats was conducted to establish the safety profile of PBD. To our knowledge, this is the first study to combine spray-drying optimization, sensory evaluation, and long-term toxicity assessment for a Thai poly-herbal beverage. These findings highlight the potential of PBD as a scientifically validated, antioxidant-rich, and safe functional beverage for commercialization in the complementary and integrative medicine sector.

## 2.  Materials and methods

### 2.1.  Preparation of *Phyllanthus emblica*-based functional herbal tea

The medicinal plants for the herbal tea (see S1 Table) were sourced from licensed supplier Triburi Orsot in Songkhla, Thailand. Each specimen was authenticated by Assistant Professor Dr. Katesarin Maneenoon, a botanist from Prince of Songkla University, using reference samples from the Materia Medica collection. The tea formulations were prepared at the Traditional Thai Medicine Hospital in Hat Yai, Thailand. After cleaning, drying, and milling, the plant parts were combined to create five distinct formulations. Portions of 2.5 g each were packaged in rectangular tea bags (5 × 8 cm2) and stored in sterilized amber glass containers at room temperature until use. To prepare tea infusions, each tea bag was steeped in 120 mL of freshly boiled water (98 ± 2°C) and brewed for precisely 3 minutes without agitation [16]. Sensory evaluation of the herbal tea was performed by a trained panel of 30 individuals at the Department of Food Sciences and Nutrition, Prince of Songkla University. The sensory assessment included parameters such as color, aroma, taste, and overall acceptability, rated on a 9-point hedonic scale ranging from 1 (extremely disliked) to 9 (extremely liked) [16].

### 2.2.  Development of microcapsules containing PBO decoction and its phenolic-rich extract

Tea infusions, prepared as mentioned above, were freeze-dried (U535-86, New Brunswick, UK), with extraction yields calculated for each formulation. The extracts obtained were stored in sterilized amber tubes at −20°C for up to three days prior to preparation for analysis (Fig 1).

 The stepwise solvent extraction method employed here follows established protocols for phenolic isolation [20,21], where initial basification (pH 8–10) enhances solubility of non-phenolics, while subsequent acidification (pH 3–4) facilitates phenolic partitioning into ethyl acetate. This approach was validated in prior studies [17,18] for optimizing polyphenol yield from botanical extracts. An aliquot (1 mL) of the PBO solution (0.2 g/mL in DMSO) was diluted with water to 20 µg/mL and mixed with 50 mL of diethyl ether in a separatory funnel. The diethyl ether fraction was collected, while the aqueous fraction was retained. The pH of the aqueous portion was adjusted to 8–10 with 20 g/100 mL $NaHCO_3$ and extracted with 50 mL of chloroform to obtain the nonphenolic fraction. This aqueous fraction was mixed with 6 mol/L HCl to adjust the pH to 3–4 and then partitioned with 50 mL of ethyl acetate. The ethyl acetate layer and the aqueous portion yielded the polyphenolic and residual fractions, respectively (Fig 1). Each partitioning process was conducted twice, with the solvents subsequently evaporated using a rotary evaporator (EZ-2 Series, Genevac Ltd., United Kingdom). The powdered samples were then stored in airtight containers at −20 °C for future use.

### 2.3.  Preparation of microcapsules from PBO decoction and its phenolic-rich extract

Microencapsulation of PBO decoction and its phenolic-rich extract was conducted using a laboratory-scale spray dryer (Buchi Mini Spray Dryer B-290, Switzerland) with a 0.7 mm nozzle aperture, as summarized in Fig 1, lower panel. Each coating material was prepared by dispersing maltodextrin (MD) and gum Arabic (GA) in the desired ratio in distilled water at a 1:5 (w/w) ratio. The mixture was allowed to swell overnight, while magnetic stirring was employed to ensure thorough

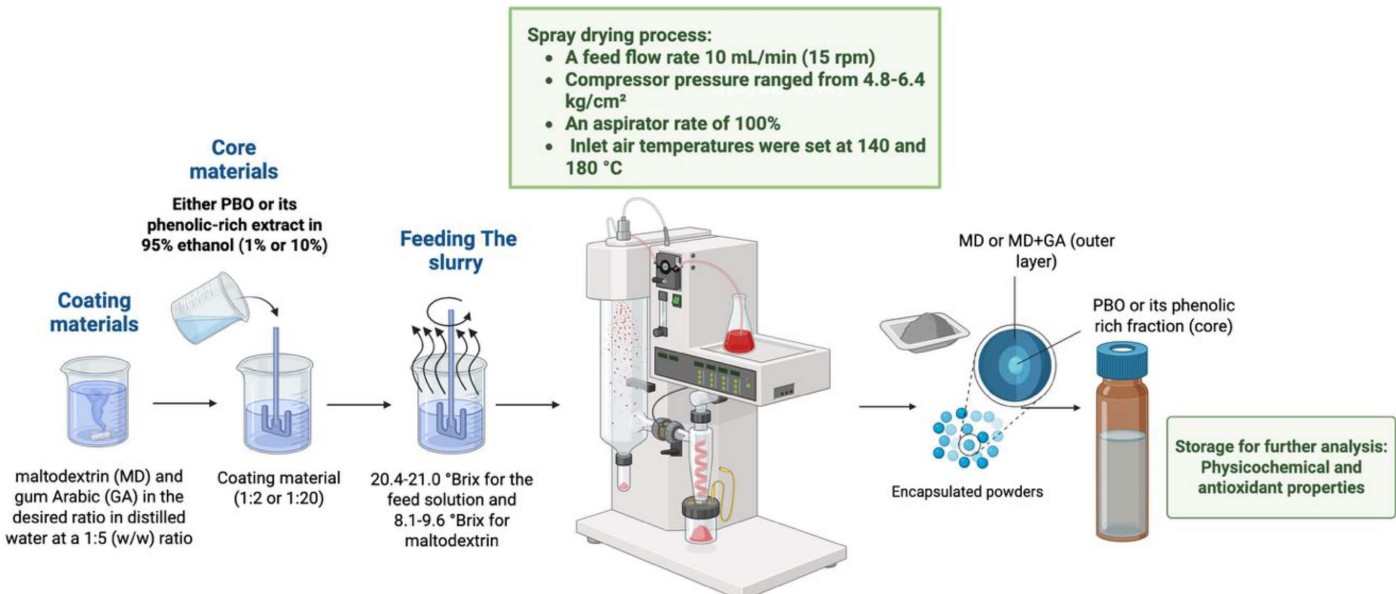

**Fig 1. Schematic workflow illustrating the preparation of Phy-Blica-O decoction (Section 2.1), extraction of its phenolic-rich extract (Section 2.2), and development of its microencapsulation using spray drying (Section 2.3).** The process starts with preparing PBO decoction, followed by freeze-drying to produce powdered extract. The phenolic-rich extract is then obtained through a series of solvent partitioning steps involving diethyl ether, chloroform, and ethyl acetate under pH-controlled conditions (upper panel). Both the PBO decoction and its phenolic-rich extract are used as core materials for microencapsulation. Coating materials, including maltodextrin (MD) and gum Arabic (GA), are dissolved in water and mixed with the core materials at different ratios. The feed slurry is spray-dried at inlet temperatures of 140°C and 180°C to produce encapsulated powders, which are then evaluated for their physicochemical and antioxidant properties (lower panel).

dissolution before the spray drying process. The feed solution was prepared by infusing 450 mL of either PBO or its phenolic-rich extract at concentrations of 1% and 10% (w/v) in 95% food-grade ethanol with the mixture of MD: GA solution through vigorous magnetic stirring at a 1:1 ratio. The feeding solutions were archived with different core-to-coating material ratios of 1:2 and 1:20 [17–19].

A total volume of 100 mL of solution was homogenized at 13500 rpm for 3 minutes using an Ultra-Turrax apparatus (IKA-Werke GmbH & Co. KG, Staufen, Germany). Soluble solid concentrations were measured at 20.4–21.0 °Brix for the feed solution and 8.1–9.6 °Brix for maltodextrin. The slurry was introduced into the main chamber via a peristaltic pump at a feed flow rate of about 10 mL/min (15 rpm). Compressor pressure ranged from 4.8–6.4 kg/cm2, with an aspirator rate of 100%. Inlet air temperatures were set at 140 and 180 °C, which were chosen based on previous studies on spray-drying of herbal extracts [13,17,18], with outlet temperatures recorded for each sample. The drying procedure was performed in triplicate. The resulting powder was collected from the cyclone chamber and stored at 4 °C in a sealed amber bottle. The weights of the microencapsulated powders were recorded, and yield efficiencies were calculated as the ratio of the total solids in the powder to those in the feed mixture.

## 2.4. Physicochemical characters of encapsulated powders

The physicochemical characteristics of encapsulated powders of PBO decoction and its phenolic-rich extract were evaluated through a series of analyses, as previously outlined in an earlier publication [17,18]. Particle size distribution and mean particle size were assessed in triplicate with a Malvern Scirocco 2000 particle analyzer (Malvern Instruments, Worcestershire, U.K.) utilizing isopropanol for dispersion. Morphological characteristics, such as surface appearance and particle size, were analyzed using scanning electron microscopy (SEM; SEMJSM5800LV model, Japan), with gold-coated samples examined at a magnification of 5000× under a 30 kV accelerating voltage. Tap density was derived by placing 2 g of powder into a 10 mL graduated cylinder and tapping until a constant volume was reached, with density calculated as the mass-to-volume ratio. Hygroscopicity was evaluated by measuring mass changes after 24 hours and 7 days of exposure to controlled humidity conditions (76.5–78% RH). Color properties ($L*$, $a*$, $b*$) were recorded using a calibrated Hunter Lab Color Quest XE colorimeter (Mini Scan X.E., Model 45/0S, Hunter Associates Laboratory Inc., USA), with measurements taken in triplicate. Moisture content was obtained through oven drying at 105°C until a consistent weight was achieved, while water activity was assessed at equilibrium using an Aqua Lab analyzer (Aqua Lab, Series 3 C.E., USA) at 25°C.

## 2.5. Antioxidant capacities of encapsulated powders and *Phyllanthus emblica*-based functional herbal tea

Encapsulated powders from PBO decoction and its phenolic-rich extract, created using different inlet temperatures and various wall material ratios, were evaluated and compared for their *in vitro* antioxidant capacities. Their radical scavenging abilities were evaluated using modified DPPH (1,1-diphenyl-2-picrylhydrazyl) and ABTS (2,20-azino-bis-3-ethylbenzthiazoline-6-sulfonic acid) assays. Encapsulated powders were initially dissolved in 50% dimethyl sulfoxide (DMSO), followed by serial two-fold dilutions to produce a concentration series from 2500 to 1.22 µg/mL. Trolox, which serves as a standard antioxidant, was also diluted (1250–0.61 µg/mL). The superoxide radical scavenging activity of each microcapsule powder was assessed using a modified nitroblue tetrazolium (NBT) reduction method. In this procedure, superoxide radicals generated by a riboflavin/methionine/illumination reaction convert NBT into purple formazan, which is quantified using a spectrophotometer. Specifically, reaction mixtures of riboflavin, methionine, EDTA, and a range of extract concentrations (156.25–4.88 µg/mL in phosphate buffer, pH 7.4) were combined with NBT solution and exposed to fluorescent light (20 W) at 25°C for 25 minutes. The absorbance of the resulting formazan dye was measured at 560 nm, with catechin being used as the reference antioxidant. Free radical inhibition was reported in terms of $IC_{50}$ (mg/mL), which represents the concentration needed to achieve 50% radical inhibition, as described by previous methods [7,9,10].

The FRAP assay was employed to measure the reducing capability of the powders. A 30 μL sample was combined with 270 μL freshly prepared FRAP reagent, and absorbance changes were recorded at 596 nm after 30 minutes of incubation. Results were expressed as millimolar equivalents of $FeSO_4$ per gram extract, determined using a standard $FeSO_4$ calibration curve. Total phenolic content was measured using the Folin-Ciocalteu method. Fifteen μL of extracts (2.5 mg/mL) reacted with 125 μL of Folin-Ciocalteu reagent for 5 minutes, then 125 μL of sodium carbonate (20% w/v) was added. After incubating in darkness for one hour, absorbance at 725 nm was recorded. TPC values were based on a gallic acid standard curve and expressed as mg GAE per gram of extract [7,9].

In addition, *Phyllanthus emblica*-based functional herbal tea decoction, as previously described, was freeze-dried, with extraction yields calculated for each formulation. Extracts obtained were stored in sterilized amber tubes at −20°C for up to three days prior to analysis. The scavenging activities against superoxide anion and peroxyl radicals were measured using NBT and the ORAC assays [7]. The lipid peroxidation inhibitory capacities of the extracts were evaluated with slight modifications to a previously established method [20]. Specifically, varying concentrations (0.1–1 mg/mL) of each extract were prepared. Ten microliters of each concentration were added to a glass tube containing 12.5 mg/mL egg yolk (50 μL), 20% acetic acid (150 μL), 0.8% TBA (150 μL), and 40 μL distilled water. The mixture was vortexed for 10 seconds and heated at 95 °C for one hour. Absorbance (532 nm) of the resulting solution was measured against a blank without egg yolk, using butylated hydroxytoluene (BHT) as a positive control.

## 2.6. Sub-chronic oral toxicity study

Healthy adult male Wistar rats (300–350 g) and female Wistar rats (200–250 g) were obtained from the National Laboratory Animal Center in Thailand. The animals were housed individually in a controlled environment with temperatures maintained between 23 and 25°C, humidity levels of 50–55%, and a 12-hour light-dark cycle. They were given sterilized feed and purified water ad libitum. The study followed OECD Guideline 408 [21–23] and received approval from the Animal Ethical Committee at Prince of Songkla University 0521.11/711).

After a two-week acclimatization, rats were randomly assigned to four groups (n = 10 males, n = 10 females per group), maintaining weight variations within ±5% between groups. The control group received distilled water orally, while the experimental groups received daily doses of Phy-Blica-D extracts at 5, 50, and 300 mg/kg body weight for 90 days.

Humane endpoints were established before the study. Animals were monitored twice daily for pain, distress, infection, or adverse reactions. Humane euthanasia criteria included persistent weight loss (>20%), decreased mobility, excessive grooming or vocalization, abnormal posture, and lack of response to stimuli. Animals showing these signs were euthanized immediately with sodium pentobarbital (≥150 mg/kg, intraperitoneal). No unexpected mortalities occurred; all animals survived until their endpoints. Animals were monitored daily for mortality, clinical symptoms, behavioral changes, signs of pain, distress, or abnormal behavior. Any animals showing signs of suffering were evaluated by a veterinarian for early humane euthanasia if necessary. Body weight and food consumption were logged daily. On the 90th day, the animals were anesthetized under deep anesthesia using an intraperitoneal injection of sodium pentobarbital (60 mg/kg), and blood samples were drawn for hematological and biochemical analyses. Additionally, organ weights were recorded, and standard histological procedures were used to conduct histopathological examinations of the liver, kidneys, lungs, and spleen.

## 2.7. Statistical analysis

Data is shown as mean ± standard error or mean ± standard deviation. Antioxidant capacity, body weights, food intake, relative organ weights, and both hematological and biochemical parameters were all assumed to follow a normal distribution. Sensory scores and group differences were examined using one-way ANOVA with Bonferroni's post-hoc tests, conducted via the Statistical Package for the Social Sciences (SPSS 19) on Windows. A P-value of less than 0.05 was deemed statistically significant.

## 3. Results

### 3.1. Microencapsulation conditions and efficiency

The selection of coating materials and the specifications regarding inlet temperatures were derived from prior research that developed and employed herbal-based microcapsules as functional ingredients for food and beverage applications. These microcapsules consist of either a crude decoction of PBO or its phenolic-rich extract as core materials. The decoction of PBO, consisting of 11 medicinal plants (S1 Table), was prepared using boiling water, following the traditional method detailed previously. The yield of this decoction was found to be 14.56% (w/w), while the phenolic-rich extract (PRF) was produced subsequently with a yield of 9.59% (w/w).

Table 1 displays the yields and efficiencies of encapsulation for crude and its PRF at different inlet temperatures (140°C, 180°C) and varying ratios of coating materials (maltodextrin [MD]: gum Arabic [GA]). Both crude extract (1:2 core-to-coating ratio, w/w) and the PRF (1:20 core-to-coating ratio, w/w) were assessed to determine optimal encapsulation parameters.

The encapsulation yield for the crude extract varied between 67.47% and 75.00%, peaking at 75.00% with an inlet temperature of 180°C and a MD:GA ratio of 6:4 (CE180_6:4). In contrast, the highest encapsulation efficiency for crude extracts was recorded at 140°C with only MD, reaching 94.03% (CE140_10:0). The lowest efficiency (75.90%) occurred at the same temperature with an MD:GA ratio of 6:4 (CE140_6:4). This clearly demonstrates that a higher MD proportion at lower temperatures improved encapsulation efficiency.

In a similar manner, the yields for the PRF ranged from 67.32% to 77.23%. The peak yield of 77.23% was obtained at 140°C with an MD to GA ratio of 8:2 (PRF140_8:2). Interestingly, the highest encapsulation efficiency for the PRF was found to be 96.29% at 140°C when using MD exclusively (PRF140_10:0), while the lowest efficiency of 70.68% occurred at 180°C with an MD:GA ratio of 6:4 (PRF180_6:4).

It is observed that a higher MD proportion (10:0) at 140°C achieves the best encapsulation efficiency (94.03% for crude extract; 96.29% for PRF), despite slightly lower results than other conditions. Additionally, increasing GA reduces microencapsulation efficiency, indicating that maltodextrin alone better preserves bioactive components during encapsulation.

**Table 1. Encapsulation conditions, yields, and efficiency of Phy-Blica-O particles prepared at various inlet temperatures and different proportions of wall materials.**

| Core material (Core: Coating material) | Codes of samples | Inlet temp. (°C) | Coating material (MD: GA) | Yield (%) | Microcapsule efficiency (%) |
|---|---|---|---|---|---|
| Crude extract (1:2; w/w) | CE140_10:0 | 140 | 10:0 | 67.47 | 94.03±0.52[a] |
| | CE140_8:2 | 140 | 8:2 | 74.69 | 86.45±0.50[c] |
| | CE140_6:4 | 140 | 6:4 | 70.71 | 75.90±0.07[e] |
| | CE180_10:0 | 180 | 10:0 | 69.41 | 89.72±0.88[b] |
| | CE180_8:2 | 180 | 8:2 | 70.85 | 84.44±1.46[d] |
| | CE180_6:4 | 180 | 6:4 | 75.00 | 84.37±0.33[d] |
| Phenolic rich fraction (1:20; w/w) | PRF140_10:0 | 140 | 10:0 | 67.32 | 96.29±0.35[a] |
| | PRF140_8:2 | 140 | 8:2 | 77.23 | 70.96±0.30[e] |
| | PRF140_6:4 | 140 | 6:4 | 72.39 | 77.98±1.20[c] |
| | PRF180_10:0 | 180 | 10:0 | 71.29 | 83.55±0.10[b] |
| | PRF180_8:2 | 180 | 8:2 | 70.63 | 73.58±0.66[d] |
| | PRF180_6:4 | 180 | 6:4 | 70.91 | 70.68±0.51[e] |

[a–e]Values are presented as mean±SD. Means within a column for each sample sharing a common superscript letter are not significantly different ($p < 0.05$) as analyzed by one-way ANOVA followed by Bonferroni's post-hoc comparison tests. MD; maltodextrin: GA; gum Arabic. The effect size ($\eta2$) for microcapsule efficiency, with 95% confidence intervals, is 0.995 (0.80–0.82).

### 3.2. Physicochemical properties of microencapsulated PBO

The physicochemical characterization of microencapsulated Phy-Blica-O powders is detailed in Table 2. It indicates the variation in particle size, bulk and tap densities, moisture content, water activity, and hygroscopicity under different encapsulation conditions. Powders encapsulated at higher inlet temperatures (180°C) showed lower moisture content and water activity but increased hygroscopicity. The smallest particle sizes and higher hygroscopicity after 7 days were notably observed with higher GA proportions, indicating potential moisture absorption issues affecting storage stability.

The average particle size of the microcapsules varied from 6.01 to 8.54 µm, exhibiting broad particle-size distributions ranging from as small as 0.13 µm to as large as 76.42 µm (see S1 and S2 Figs). The bulk density values ranged from 0.20 to 0.25 kg/m2, while tap densities spanned 0.35 to 0.45 kg/m2. This indicates moderate variability among formulations, which could affect handling and storage. Moisture content was notably affected by drying conditions, showing lower moisture levels (4.21%) at elevated temperatures (180°C, PRF180_8:2), whereas lower temperatures resulted in higher moisture content (up to 7.11%, CE140_8:2). Water activity also fluctuated, with higher activity (up to 0.37, CE140_8:2 and PRF140_6:4) linked to the lower drying temperatures. Hygroscopicity tests revealed greater moisture absorption during storage. Formulations with higher gum arabic ratios, especially CE180_6:4, exhibited the highest hygroscopicity (15.51% after 7 days), indicating potential concerns regarding storage and stability in humid environments.

Effect sizes ($\eta2$) and 95% confidence intervals (CI) were calculated to complement p-values and quantify the magnitude of treatment effects. Microencapsulation efficiency demonstrated an extremely large effect ($\eta2 = 0.995$, 95% CI: 0.80–0.82). Significant differences in physicochemical properties were also associated with large effect sizes, including moisture content ($\eta2 = 0.946$, 95% CI: 0.84–0.97), water activity ($\eta2 = 0.940$, 95% CI: 0.82–0.96), and hygroscopicity after 7 days ($\eta2 = 0.859$, 95% CI: 0.70–0.93). These results indicate that inlet temperature and wall-material ratios have strong practical effects on powder characteristics.

Overall, these physicochemical evaluations underscore the significant effects of drying temperatures and wall material compositions on the characteristics of microcapsules, highlighting the need for optimized parameters to enhance the stability and quality of functional microencapsulated products.

**Table 2. The physicochemical properties of microencapsulated Phy-Blica-O produced at different inlet temperatures and with varying proportions of wall materials.**

| Samples | Average particle size (µm) (Range) | Bulk density (Kg/m²) | Tap density (Kg/m²) | Moisture content (%) | Water activity (%) | Hygroscopicity (%) | |
|---|---|---|---|---|---|---|---|
| | | | | | | 24 hours | 7 days |
| CE140_10:0 | 6.67±4.03 (0.26-22.73) | 0.23±0.00$^b$ | 0.39+0.00$^d$ | 6.26±0.03$^c$ | 0.35±0.00$^b$ | 10.32±0.07$^d$ | 11.22±0.08$^e$ |
| CE140_8:2 | 7.92±5.80 (0.21-76.42) | 0.23±0.00$^b$ | 0.39±0.01$^{c,d}$ | 7.11±0.02$^a$ | 0.37±0.00$^a$ | 12.27±0.07$^c$ | 12.99±0.06$^d$ |
| CE140_6:4 | 6.01±3.74 (0.21-22.73) | 0.25±0.00$^a$ | 0.39±0.00$^{b,c,d}$ | 6.47±0.03$^b$ | 0.33±0.00$^c$ | 10.10±0.06$^e$ | 11.02±0.05$^f$ |
| CE180_10:0 | 6.78±4.16 (0.26-24.95) | 0.21±0.00$^d$ | 0.40±0.00$^{b,c}$ | 4.51±0.15$^e$ | 0.25±0.00$^e$ | 12.83±0.06$^b$ | 13.89±0.06$^b$ |
| CE180_8:2 | 6.93±4.28 (0.24-27.39) | 0.23±0.00$^b$ | 0.40±0.00$^b$ | 5.03±0.09$^d$ | 0.26±0.00$^d$ | 12.34±0.07$^c$ | 13.10±0.05$^c$ |
| CE180_6:4 | 6.34±4.06 (0.24-43.67) | 0.22±0.00$^c$ | 0.45±0.01$^a$ | 4.96±0.08$^d$ | 0.23±0.00$^f$ | 14.29±0.07$^a$ | 15.51±0.08$^a$ |
| PRF140_10:0 | 8.54±5.88 (0.26-63.41) | 0.22±0.00$^b$ | 0.35±0.01$^e$ | 5.44±0.01$^b$ | 0.34±0.00$^b$ | 10.21±0.02$^f$ | 10.70±0.02$^f$ |
| PRF140_8:2 | 7.36±4.21 (0.16-24.95) | 0.22±0.00$^b$ | 0.39±0.01$^c$ | 5.28±0.06$^c$ | 0.31±0.00$^c$ | 11.53±0.07$^c$ | 11.95±0.06$^c$ |
| PRF140_6:4 | 7.86±4.50 (0.16-30.07) | 0.20±0.00$^c$ | 0.41±0.01$^b$ | 6.38±0.02$^a$ | 0.37±0.00$^a$ | 12.23±0.03$^b$ | 12.51±0.07$^b$ |
| PRF180_10:0 | 8.08±4.82 (0.21-30.07) | 0.22±0.00$^b$ | 0.36±0.01$^d$ | 4.85±0.04$^d$ | 0.30±0.00$^d$ | 11.27±0.08$^d$ | 11.56±0.07$^d$ |
| PRF180_8:2 | 6.79±3.76 (0.13-22.73) | 0.20±0.00$^c$ | 0.41±0.00$^b$ | 4.21±0.01$^e$ | 0.24±0.00$^f$ | 11.05±0.08$^e$ | 11.10±0.09$^e$ |
| PRF180_6:4 | 6.89±3.68 (0.18-20.71) | 0.24±0.00$^a$ | 0.44±0.01$^a$ | 4.89±0.07$^d$ | 0.26±0.00$^e$ | 12.59±0.03$^a$ | 12.95±0.03$^a$ |

$^{a–f}$Values are presented as mean±SD. Means within each column that share the same superscript letter indicate no significant difference (p<0.05), determined by one-way ANOVA followed by Bonferroni's post-hoc tests. The effect size ($\eta2$) for moisture content, water activity, and hygroscopicity at 7 days, with 95% confidence intervals, is 0.946 (0.84–0.97), 0.940 (0.82–0.96), and 0.859 (0.70–0.93), respectively.

## 3.3. Appearance and color parameters of microencapsulated PBO powders

Color properties (Table 3), measured as *L\** (lightness), *a\** (redness-greenness), *b\** (yellowness-blueness), Delta E (color difference), and whiteness values, are crucial for determining consumer acceptance and market potential of functional powders. Typically, a rise in GA proportion led to a decrease in whiteness and an increase in yellowness (higher b\* values), which could impact visual acceptability. Additionally, phenolic-rich fractions encapsulated at elevated temperatures showed enhanced whiteness, hinting at a potential benefit for consumer preference when choosing encapsulation conditions. Microcapsules produced through spray drying presented spherical forms with varying surface indentations and particle integrity, influenced by the inlet temperatures (140°C and 180°C) and the MD to GA ratio (Figs 2 and 3). At both temperatures, microcapsules made solely with MD exhibited relatively consistent spherical shapes with minor surface indentations, indicating satisfactory encapsulation integrity. An increase in the GA proportion led to microcapsules featuring more pronounced dents, irregular surfaces, and greater size variability, especially notable at the 6:4 MD:GA ratio. Furthermore, samples dried at the higher temperature of 180°C appeared smoother and more compact compared to those dried at 140°C, suggesting enhanced structural stability and possible benefits for the product's shelf-life. These findings emphasize the crucial role of formulation parameters in determining the physical quality and integrity of microencapsulated powders.

## 3.4. Antioxidant properties of microencapsulated PBO powders

The antioxidant evaluations presented in Table 4 reveal differences in antioxidant strength depending on encapsulation conditions. Microcapsules containing higher proportions of GA (CE140_6:4 and CE180_6:4) exhibited significantly elevated total phenolic content and strong antioxidant activity, as evidenced by the lowest $IC_{50}$ values in the ABTS assay, highlighting the protective influence of GA on phenolic compounds. The phenolic-rich fractions displayed significantly lower antioxidant activities due to the reduced amounts of core material used in the preparation of microcapsules. This suggests that crude extracts could be suitable for creating microcapsules with higher antioxidant strength. The total phenolic content in crude extracts ranged from 85.88 to 102.05 mg GA/g extract, which is notably higher than that in phenolic-rich fractions (10.55 to 17.77 mg GA/g extract). Among these crude extracts, CE140_6:4 had the highest total phenolic content (102.05 mg GA/g extract), closely followed by CE180_6:4 (101.71 mg GA/g extract). In radical

**Table 3. The appearance of microencapsulated Phy-Blica-O powders varies based on different inlet temperatures and wall material proportions.**

| Codes | L* | a* | b* | Delta E | Whiteness value |
|---|---|---|---|---|---|
| CE140_10:0 | 80.04±1.02[b] | 1.74±0.10[a] | 19.17±0.63[a] | 24.03±1.06[b] | 72.27±1.10[b] |
| CE140_8:2 | 82.08±1.07[a] | 1.15±0.07[c] | 17.91±0.52[d] | 21.76±1.06[c] | 74.64±1.12[a] |
| CE140_6:4 | 78.45±0.77[c] | 1.68±0.09[b] | 19.34±0.34[a] | 25.19±0.68[a] | 70.99±0.72[c] |
| CE180_10:0 | 80.19±1.24[b] | 1.65±0.10[b] | 18.21±0.54[c] | 23.20±1.11[c] | 73.03±1.18[b] |
| CE180_8:2 | 81.51±2.16[a] | 1.15±0.06[c] | 17.52±0.59[e] | 21.83±1.78[c] | 74.48±1.91[a] |
| CE180_6:4 | 78.48±1.14[c] | 1.72±0.12[a] | 18.75±0.56[b] | 24.74±1.01[a,b] | 71.40±1.06[c] |
| PRF140_10:0 | 89.24±1.54[c,d] | 0.16±0.03[c] | 9.83±0.24[b] | 11.07±1.00[a] | 85.39±1.23[c] |
| PRF140_8:2 | 88.97±1.26[c,d] | 0.44±0.04[a] | 10.02±0.48[b] | 11.35±1.06[a] | 85.08±1.23[c] |
| PRF140_6:4 | 89.67±1.41[c] | 0.15±0.05[c] | 10.28±0.43[a] | 11.23±0.94[a] | 85.40±1.18[c] |
| PRF180_10:0 | 90.43±0.95[b] | 0.24±0.03[b] | 8.96±0.32[d] | 9.68±0.68[b] | 86.88±0.87[b] |
| PRF180_8:2 | 91.59±1.19[a] | 0.09±0.03[d] | 8.41±0.36[e] | 8.70±0.73[c] | 88.09±1.02[a] |
| PRF180_6:4 | 89.14±1.07[c,e] | 0.15±0.04[c] | 9.42±0.40[c] | 10.75±0.80[a] | 85.61±0.95[c] |

[a–e]Values are expressed as mean±SD (n=9). Means within a column for each sample sharing a different superscript letter indicate a significant difference (p<0.05), based on one-way ANOVA followed by Bonferroni's post-hoc comparison tests.

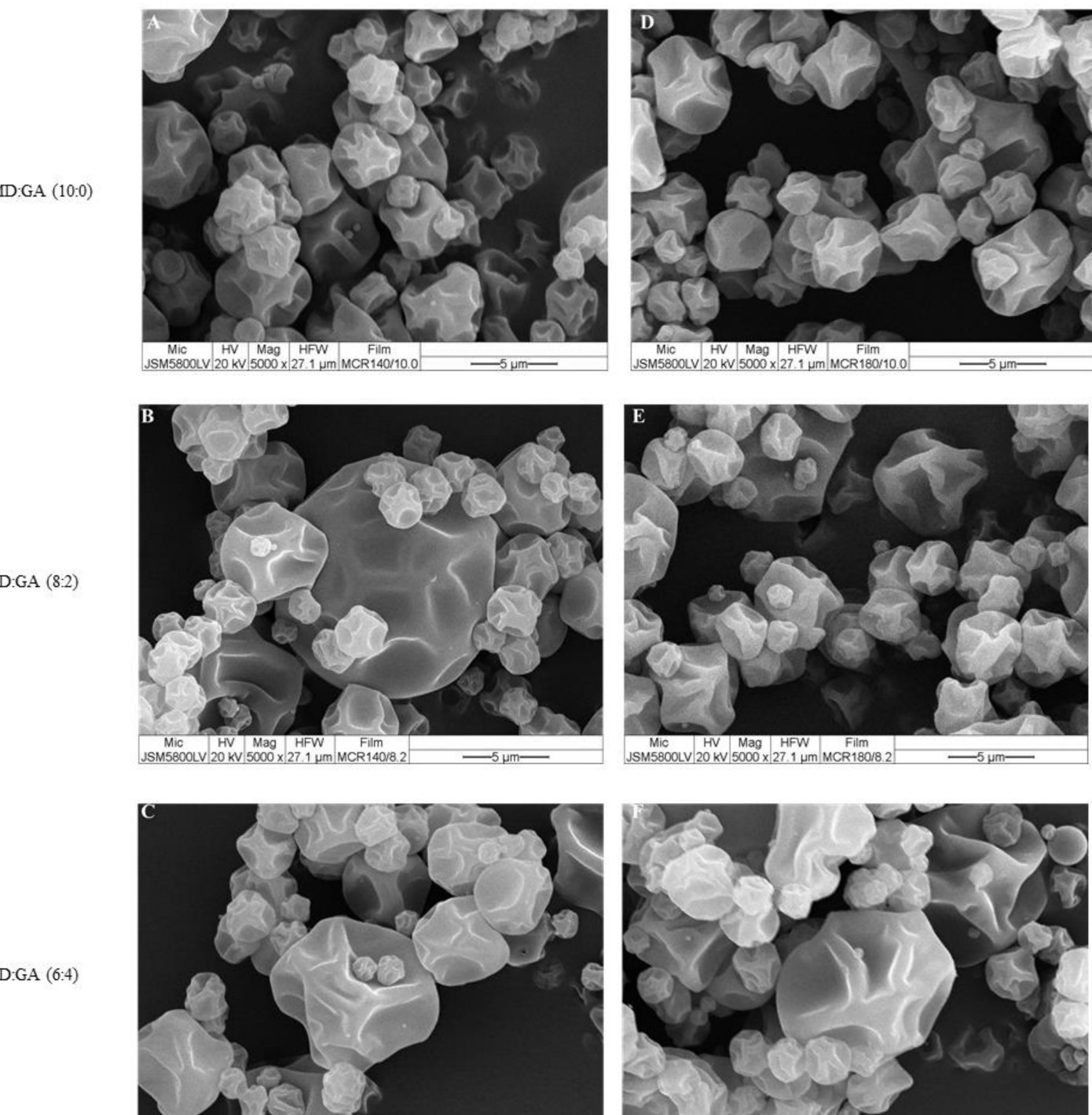

**Fig 2. Scanning electron micrographs depict the morphology of microencapsulated Phy-Blica-O powders created through spray drying at inlet temperatures of 140°C (A–C) and 180°C (D–F).** The powders utilize different maltodextrin (MD) to gum Arabic (GA) ratios: **(A, D)** MD:GA (10:0), **(B, E)** MD:GA (8:2), and **(C, F)** MD:GA (6:4). The scale bars indicate 5 μm with a magnification of ×5000.

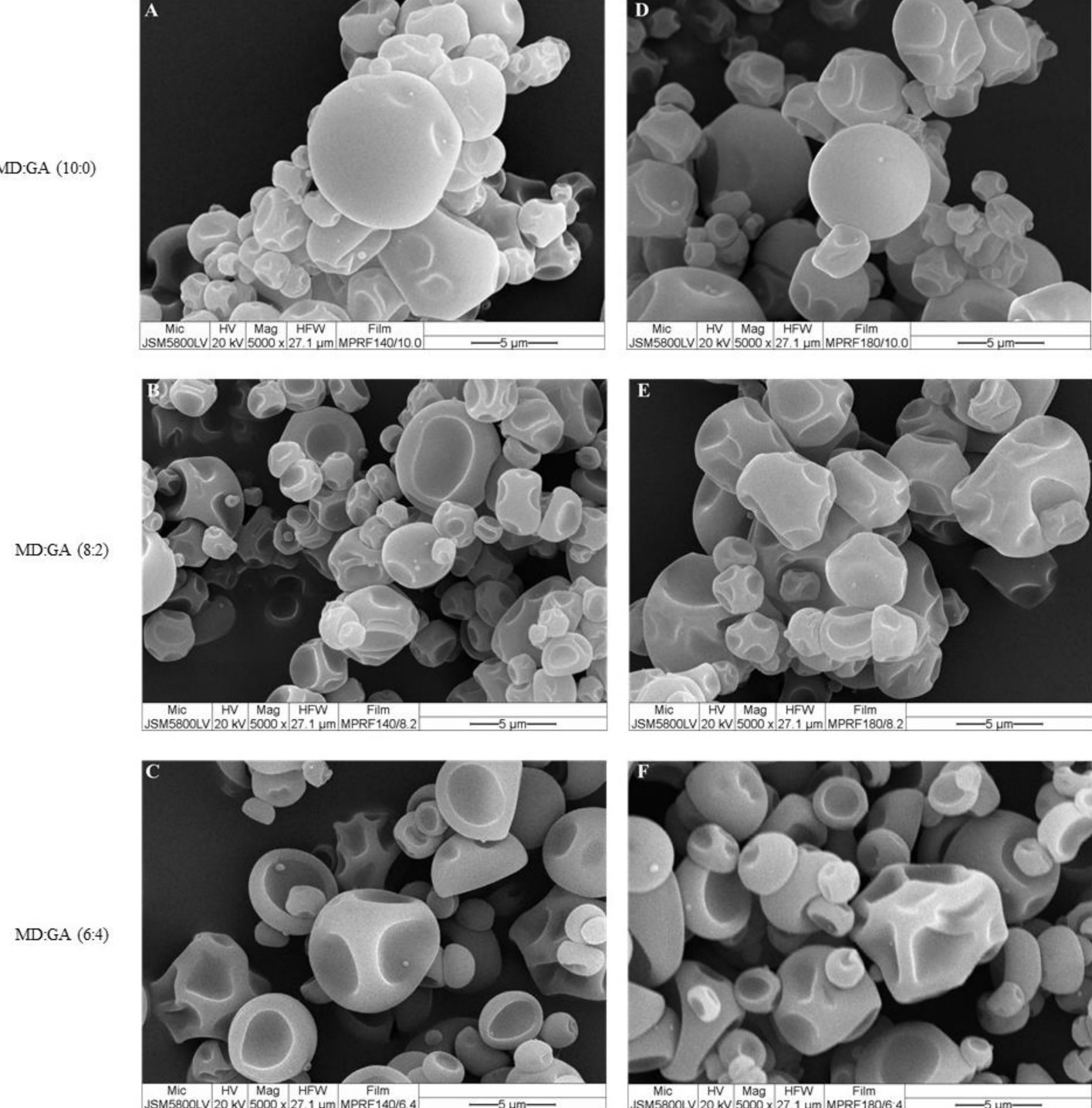

**Fig 3. Scanning electron micrographs depict the morphology of microencapsulated phenolic-rich fraction of Phy-Blica-O powders created through spray drying at inlet temperatures of 140°C (A–C) and 180°C (D–F).** The powders utilize different maltodextrin (MD) to gum Arabic (GA) ratios: **(A, D)** MD:GA (10:0), **(B, E)** MD:GA (8:2), and **(C, F)** MD:GA (6:4). The scale bars indicate 5 μm with a magnification of ×5000.

**Table 4. The antioxidant properties of microencapsulated Phy-Blica-O and its phenolic-rich fraction, produced with varying inlet temperatures and different proportions of wall materials.**

| Codes | Total phenolic content mg GA/g extract | DPPH ($IC_{50}$) mg/mL | ABTS ($IC_{50}$) mg/mL | FRAP mM $Fe_2SO_4$/mg | NBT ($IC_{50}$) mg/mL |
|---|---|---|---|---|---|
| CE140_10:0 | 87.64±2.77[c] | 0.311±0.003[d] | 0.156±0.004[d] | 3.55±0.04[c] | 0.29±0.00[a] |
| CE140_8:2 | 85.88±1.54[c] | 0.409±0.006[b] | 0.258±0.001[e] | 3.62±0.03[c] | 0.38±0.02[b] |
| CE140_6:4 | 102.05±3.30[a] | 0.375±0.016[c] | 0.109±0.003[b] | 3.82±0.01[b] | 0.51±0.03[d] |
| CE180_10:0 | 87.44±2.88[c] | 0.515±0.005[a] | 0.162±0.003[d] | 3.76±0.02[b] | 0.29±0.01[a] |
| CE180_8:2 | 93.25±2.13[b] | 0.422±0.004[b] | 0.129±0.005[c] | 4.10±0.02[a] | 0.37±0.01[b] |
| CE180_6:4 | 101.71±3.25[a] | 0.378±0.017[c] | 0.095±0.003[a] | 4.15±0.08[a] | 0.41±0.01[c] |
| PRF140_10:0 | 14.62±0.87[c] | 1.238±0.015[e] | 1.323±0.007[c] | 0.42±0.01[d] | 1.58±0.06[b] |
| PRF140_8:2 | 10.88±0.45[e] | 2.211±0.004[b] | 1.536±0.003[d] | 0.39±0.01[e] | 1.46±0.04[a] |
| PRF140_6:4 | 16.82±0.62[b] | 2.704±0.009[a] | 1.106±0.003[b] | 0.65±0.02[b] | 1.68±0.04[c] |
| PRF140_10:0 | 10.55±0.42[e] | 1.974±0.004[c] | 1.443±0.004[c,d] | 0.50±0.01[c] | 2.23±0.10[c] |
| PRF140_8:2 | 12.27±0.71[d] | 1.865±0.002[d] | 1.359±0.001[c] | 0.74±0.00[a] | 2.34±0.11[c] |
| PRF140_6:4 | 17.77±0.97[a] | 2.717±0.008[a] | 1.021±0.002[a] | 0.67±0.01[b] | 2.64±0.03[d] |

[a–e]Values are expressed as mean±SD. Means within a column for each sample sharing a different superscript letter indicate a significant difference (p<0.05), based on one-way ANOVA followed by Bonferroni's post-hoc comparison tests.

scavenging assays, crude extracts demonstrated strong antioxidant activity, indicated by lower $IC_{50}$ values, particularly CE140_10:0 and CE180_6:4 in both the ABTS (0.156 and 0.095 mg/mL, respectively) and DPPH assays. Consistently, the FRAP assay, which measures reducing power, also showed superior activity for CE180_6:4 (4.15 mM $Fe_2SO_4$/mg).

Although CE140_6:4 exhibited the highest TPC, CE180_6:4 consistently showed superior activity in DPPH, ABTS, and FRAP assays (p<0.05). This suggests that antioxidant potency may be driven not only by TPC quantity but also by the retention and stability of specific phenolic compounds during high-temperature drying. Collectively, CE180_6:4 consistently showed superior antioxidant performance across multiple antioxidant assays, marking it as the optimal microcapsule formulation for maximizing antioxidant activity. These results emphasize the significant impact of encapsulation conditions and wall material proportions on preserving antioxidant properties, underscoring the ideal conditions for enhancing bioactivity in functional food formulations.

### 3.5. Radical scavenging capacity and lipid peroxidation inhibition of *Phyllanthus emblica*-based herbal teas

The microcapsules developed demonstrate notable antioxidant properties comparable to those of the PBO decoction. Consequently, both the microcapsule and the decoction can serve as functional beverages. Nonetheless, certain spices and herbs found in PBO, such as *Allium sativum* and *Tinospora crispa*, impart a strong bitter taste and unpleasant odor, limiting their application in the functional food industry. To enhance consumer acceptance, we incorporated various edible medicinal plants and spices into the formulation. S2 Table presents the sensory properties of *P. emblica*-based herbal tea samples made with different medicinal plants. The samples (Phy-Blica-A, Phy-Blica-B, Phy-Blica-C, Phy-Blica-D, and Phy-Blica-E) were sensorially compared to the original tea (PBO). Mean scores for odor, taste, and overall acceptability showed no significant differences among the herbal formulations, except for PBO. It should be noted that among the five formulations tested, Phy-Blica-D (PBD) scored the highest in taste (5.50±0.35; p<0.05) and overall acceptability (5.60±0.37; p<0.05), significantly outperforming PBO (2.70±0.34 and 2.90±0.36). The *P. emblica*-based tea with *Aegle marmelos* scored higher for taste, appearance, and acceptability. Panelists favored PBD (32.5%) over Phy-Blica-B (30%), Phy-Blica-E (26%), Phy-Blica-C (9%), and Phy-Blica-A (2.5%). Given the consumer acceptability results, Phy-Blica-B, Phy-Blica-D, and Phy-Blica-E were assessed for antioxidant capacity Phy-Blica-O.

Table 5 presents the scavenging activity against peroxyl and superoxide anion radicals, along with the inhibitory effects on lipid peroxidation, of four different *Phyllanthus emblica*-based herbal tea formulations (Phy-Blica-O, B, D, and E). The antioxidant capacities were assessed using peroxyl radical scavenging (expressed as µM Trolox equivalents [TE]/µg extract) and the kinetic inhibition of fluorescence decay in fluorescein (S3 Fig), superoxide anion radical scavenging ($IC_{50}$), and inhibition of lipid peroxidation (percentage inhibition of MDA). Among the tested herbal teas, PBD exhibited the highest peroxyl radical scavenging activity ($10.95 \pm 0.12$ µM TE/µg), followed by PBO ($9.22 \pm 0.09$ µM TE/µg). For superoxide anion radical scavenging, PBO demonstrated superior activity with the lowest $IC_{50}$ ($47.07 \pm 1.84$ µg/mL), indicating more substantial scavenging potential compared to other formulations. Similarly, PBO showed a notable inhibitory effect on lipid peroxidation ($77.39 \pm 6.21\%$ inhibition), markedly higher than other tested herbal teas.

## 3.6. Sub-chronic oral toxicity assessment

A 90-day sub-chronic oral toxicity assessment was conducted on the polyherbal tea PBD, which is recognized for its potent antioxidant properties and significant consumer acceptance, to evaluate its safety and identify potential health risks associated with prolonged usage. In the course of the 90-day repeated oral dose toxicity study, no treatment-related mortality was recorded. Additionally, no behavioral alterations or observable symptoms of toxicity attributed to the administration of PBD were noted in the subjects throughout the treatment duration.

During the 90-day study, both male and female rats across all treatment groups exhibited a steady, progressive increase in body weight, aligning with normal growth patterns. No unusual fluctuations, sudden drops, or dose-related deviations were noted compared to the control group. Weekly food intake also remained stable across all groups, with only minor physiological changes from week to week that stayed within expected ranges, showing no signs of treatment-related effects. These consistent trends in body weight and food consumption throughout the study support the conclusion that Phy-Blica-D administration at all tested doses did not produce adverse effects (Fig 4).

The findings indicate that the repeated oral administration of PBD (5, 50, and 300 mg/kg) did not result in any alterations to the structural morphology of the liver, kidneys, lungs, and spleen in the treated rats when compared to the control group (refer to Table 6). Furthermore, no significant variations in the relative weights of the liver, kidneys, lungs, and spleen were noted among all PBD-dosed rats in comparison to the control rats associated with both sexes.

As illustrated in Table 7, the statistical analysis of hematological parameters reveals no significant differences between the treated and control groups across both genders at the conclusion of the treatment period. Additionally, Fig 5 further demonstrates that there were no significant variations in the plasma levels of alkaline phosphatase (ALP), alanine

**Table 5. The scavenging capacity of *Phyllanthus emblica*-based functional herbal tea against peroxyl and superoxide anion radicals and its inhibitory effect on lipid peroxidation.**

| Tested extracts | Scavenging activity against | | MDA inhibition (%; mean±SD) |
|---|---|---|---|
| | Peroxyl radical | Superoxide anion radical | |
| | (µM TE/µg of extract) | $IC_{50}$ (µg/mL) | |
| Phy-Blica-O | $9.22 \pm 0.09$[b] | $47.07 \pm 1.84$[a] | $77.39 \pm 6.21$[a] |
| Phy-Blica-B | $4.89 \pm 0.19$[d] | $128.76 \pm 3.24$[c] | $43.48 \pm 1.66$[b] |
| Phy-Blica-D | $10.95 \pm 0.12$[a] | $82.63 \pm 10.08$[b] | $32.85 \pm 21.15$[c] |
| Phy-Blica-E | $6.42 \pm 0.19$[c] | $94.25 \pm 5.46$[b] | $35.51 \pm 0.69$[c] |

Regarding superoxide anion scavenging, the $IC_{50}$ value of catechin, a reference antioxidant, was measured at $6.15 \pm 0.29$. The lipid peroxidation inhibitory effect of *Phyllanthus emblica*-based functional herbal tea was evaluated at a concentration of 5 µg/mL, and the percentage of malondialdehyde (MDA) for butylated hydroxyanisole, another reference antioxidant, was recorded at $85.05 \pm 2.71\%$. [a–c] Values are expressed as mean±SD. Means within a column for each sample sharing a different superscript letter indicate a significant difference ($p < 0.05$), based on one-way ANOVA followed by Bonferroni's post-hoc comparison tests.

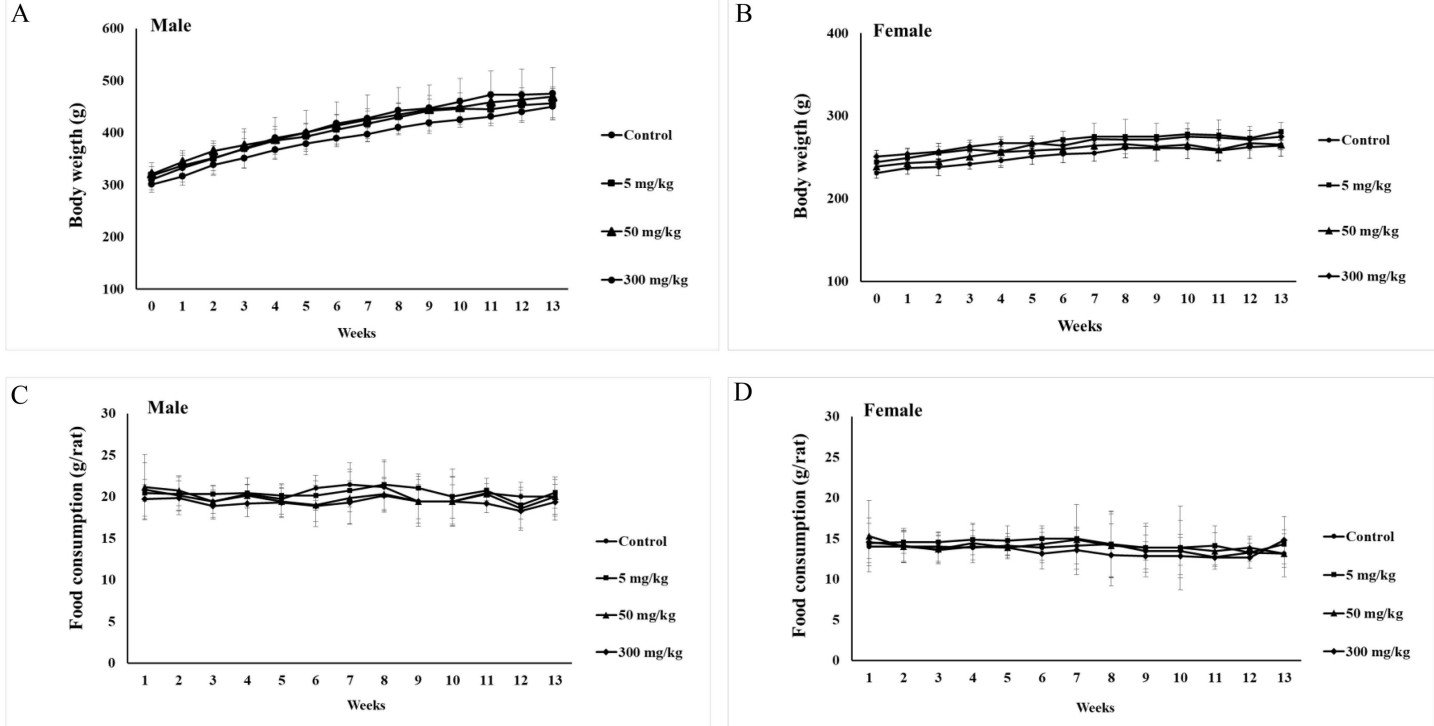

**Fig 4. Changes in body weight (panels A and B) and food consumption (panels C and D) in male and female rats exposed to doses of 5, 50, and 300 mg/kg/day during a 90-day repeated oral dose toxicity study of Phy-Blica-D are presented as mean±SD (n=5).** * Indicating a significant difference (p<0.05) among the groups each week, based on one-way ANOVA followed by Bonferroni's post-hoc comparison tests.

**Table 6. Relative organ weights of male and female rats administered oral doses of 5, 50, and 300 mg/kg/day of Phy-Blica-D extract during a 90-day repeated oral dose toxicity study.**

| Organs | Percentage of relative organ weight (mean±SD) | | | |
|---|---|---|---|---|
| | Control | Doses of Phy-Blica-D extract (mg/kg/day) | | |
| | | 5 | 50 | 300 |
| Male | | | | |
| Liver | 2.987±0.155 | 3.032±0.238 | 2.954±0.207 | 2.927±0.105 |
| Spleen | 0.196±0.022 | 0.193±0.013 | 0.181±0.017 | 0.186±0.011 |
| Kidney | 0.521±0.021 | 0.533±0.023 | 0.501±0.048 | 0.518±0.020 |
| Lung | 0.355±0.026 | 0.349±0.015 | 0.348±0.019 | 0.355±0.017 |
| Female | | | | |
| Liver | 2.893±0.177 | 2.936±0.291 | 2.914±0.271 | 2.794±0.256 |
| Spleen | 0.246±0.011 | 0.244±0.033 | 0.229±0.025 | 0.236±0.036 |
| Kidney | 0.561±0.025 | 0.578±0.024 | 0.546±0.040 | 0.552±0.023 |
| Lung | 0.439±0.037 | 0.417±0.020 | 0.448±0.018 | 0.419±0.035 |

Values are expressed as mean±SD. * Within a row for each organ, indicates a significant difference (p<0.05), based on one-way ANOVA followed by Bonferroni's post-hoc comparison tests.

**Table 7. Hematological parameters in male and female rats treated orally with doses of 5, 50, and 300 mg/kg/day of Phy-Blica-D extract during a 90-day repeated oral dose toxicity study.**

| Parameters (unit)* | Control | Dose of Phy-Blica-D extract (mg/kg/day) | | |
|---|---|---|---|---|
| | | 5 | 50 | 300 |
| *Male rats (n = 5)* | | | | |
| White blood cell count ($10^3$/mm$^3$) | 3.76±0.33 | 3.50±0.92 | 3.93±0.32 | 3.31±0.82 |
| Lymphocytes (%) | 75.28±8.51 | 76.58±3.21 | 76.23±6.63 | 77.78±6.29 |
| Monocytes (%) | 12.90±1.97 | 12.15±2.77 | 11.29±3.11 | 11.65±3.94 |
| Granulocytes (%) | 13.05±3.44 | 13.90±3.37 | 14.15±2.82 | 14.33±4.19 |
| Red blood cell count ($10^6$/mm$^3$) | 8.78±0.76 | 8.67±0.34 | 8.83±0.48 | 8.65±0.32 |
| Hemoglobin (g/dL) | 14.64±0.89 | 14.68±0.41 | 14.90±0.69 | 14.94±0.44 |
| Hematocrit (%) | 49.54±0.78 | 49.48±1.38 | 48.26±1.91 | 48.10±1.46 |
| Mean corpuscular volume (fL) | 56.44±0.94 | 57.12±1.14 | 56.76±1.92 | 55.50±1.56 |
| Mean corpuscular hemoglobin (pg) | 17.02±0.55 | 17.20±0.60 | 17.64±0.48 | 16.94±0.62 |
| Red cell distribution width (%) | 13.70±0.27 | 13.48±0.67 | 13.52±0.26 | 13.56±0.49 |
| Platelet count ($10^3$/mm$^3$) | 640.20±46.86 | 664.00±84.07 | 693.40±66.53 | 667.33±74.11 |
| *Female rats (n = 5)* | | | | |
| White blood cell count ($10^3$/mm$^3$) | 1.66±0.81 | 1.36±0.39 | 1.96±0.96 | 1.84±0.95 |
| Lymphocytes (%) | 78.10±10.75 | 76.19±6.88 | 71.39±8.35 | 73.35±8.42 |
| Monocytes (%) | 13.72±2.39 | 13.33±3.46 | 14.15±5.43 | 14.93±5.16 |
| Granulocytes (%) | 11.35±5.90 | 12.30±2.80 | 13.68±3.45 | 11.55±4.32 |
| Red blood cell count ($10^6$/mm$^3$) | 8.07±0.40 | 7.80±0.28 | 8.05±0.26 | 8.53±0.26 |
| Hemoglobin (g/dL) | 15.10±0.94 | 14.54±0.87 | 14.66±0.40 | 15.21±0.88 |
| Hematocrit (%) | 48.46±2.36 | 46.44±1.18 | 46.34±1.91 | 49.03±2.46 |
| Mean corpuscular volume (fL) | 59.02±2.87 | 59.36±1.36 | 57.62±2.18 | 57.48±2.26 |
| Mean corpuscular hemoglobin (pg) | 18.70±0.37 | 18.90±0.42 | 18.22±0.68 | 18.93±0.51 |
| Red cell distribution width (%) | 12.34±0.17 | 12.18±0.29 | 12.28±0.19 | 12.40±0.24 |
| Platelet count ($10^3$/mm$^3$) | 743.60±84.00 | 791.60±50.79 | 713.40±82.66 | 804.25±69.17 |

Values are expressed as mean±SD. * Within a row for each organ, indicates a significant difference ($p < 0.05$), based on one-way ANOVA followed by Bonferroni's post-hoc comparison tests.

aminotransferase (ALT), blood urea nitrogen (BUN), and creatinine among the treatment and control groups for both sexes of the animals that consistently received oral doses of PBD at dosages of 5, 50, and 300 mg/kg over the 28-day study duration. The histopathological examination of the vital organs, including the kidneys, liver, lungs, and spleen, exhibited no consistent treatment-related macroscopic or histological alterations in either sex of the animals administered PBD (Fig 6). The histological sections of the liver displayed normal architecture, with hepatocytes organized around central veins, revealing no evidence of necrosis, lesions, or any pathological damage. Furthermore, the observation of histological sections of the kidneys from PBD-treated rats revealed adequate glomeruli and normal tubules, with no signs of glomerular damage or the presence of lumen casts. Lastly, no histopathological lesions related to the test article were identified in the spleen, which exhibited normal histology of the white pulp, red pulp, and central arterioles.

## 4. Discussion

Traditional polyherbal formulations are gaining considerable attention in functional foods due to their health-promoting properties, minimal side effects, and potential benefits in preventing age-related and chronic illnesses [24–26]. This study highlights the possibility of developing and optimizing a traditional polyherbal beverage, PBO, as a functional ingredient utilizing an advanced microencapsulation technique.

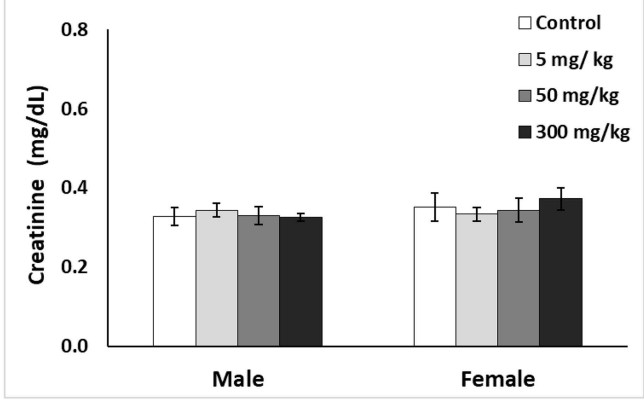
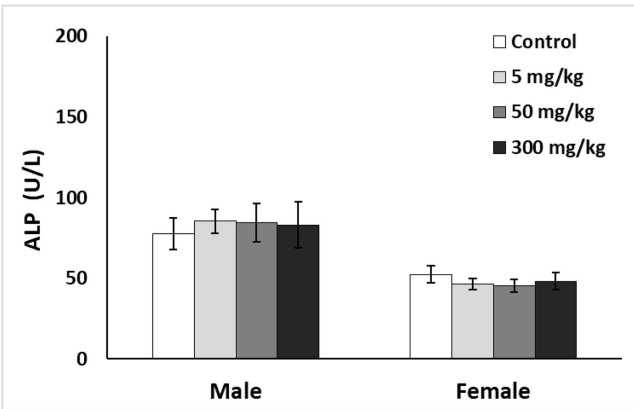
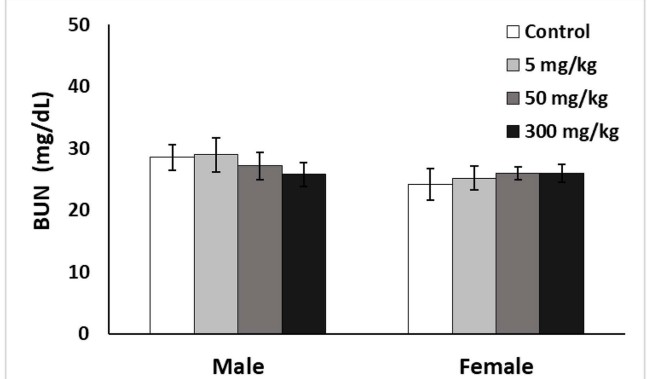
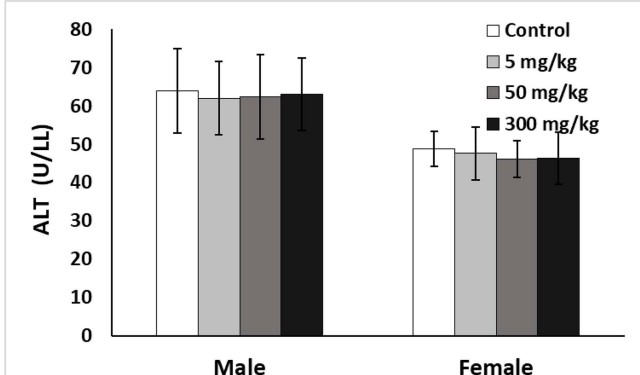

**Fig 5. Serum biochemical values of male and female rats were measured at the end of a 90-day repeated oral dose toxicity study of Phy-Blica-D extract.** Values are presented as mean ± SD (n = 5). Key parameters include creatinine, ALP (alkaline phosphatase), BUN (blood urea nitrogen), and ALT (alanine aminotransferase). * Indicating a significant difference (p < 0.05) among the groups for each parameter, based on one-way ANOVA followed by Bonferroni's post-hoc comparison tests.

Our findings indicated that encapsulation efficiency was significantly influenced by both drying temperature and the composition of encapsulation materials, specifically MD and GA. Optimal encapsulation efficiencies for the PBO and its phenolic-rich extracts were achieved at lower drying temperatures (140°C), with higher proportions of MD, consistent with previous research highlighting the effectiveness of maltodextrin in preserving bioactive components during drying processes [12–14,27–29]. This protective role is attributed to its low viscosity, high water solubility, and excellent film-forming ability, which contribute to forming a stable encapsulating matrix around heat-sensitive phytochemicals, thereby minimizing thermal degradation and oxidation during spray-drying [12,27–29]. Conversely, increased GA content led to lower encapsulation efficiencies, likely due to enhanced solution viscosity, which may impede effective drying, as observed in earlier studies [12–14].

Physicochemical analyses revealed that microcapsules produced at elevated drying temperatures had lower moisture contents and water activity levels, conditions favorable for enhancing product stability and prolonging shelf-life [29–31]. Such characteristics significantly reduce microbial contamination and chemical degradation, both critical for market acceptance and consumer safety. In addition, microcapsules containing high GA proportions, such as CE180_6:4, which showed the highest hygroscopicity (15.51% after 7 days), can lead to moisture uptake, particle caking, reduced flowability, and potential degradation of antioxidant potency during storage. These issues might impact shelf life and handling processes during manufacturing and distribution. Nevertheless, formulations with increased GA content exhibited elevated

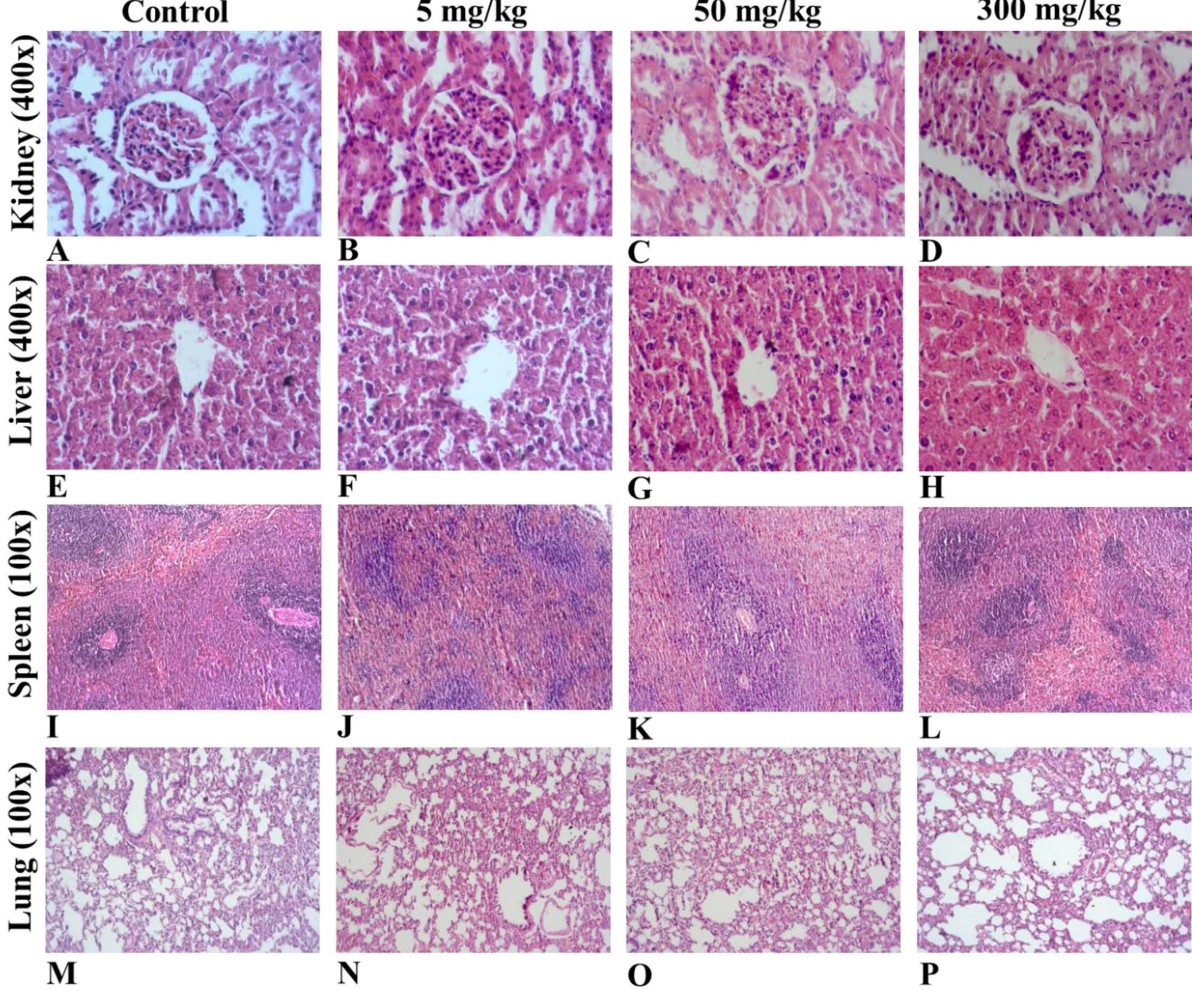

**Fig 6. Photomicrographs of kidney (A-D), liver (E-H), spleen (I-L), and lung (M-P) sections stained with hematoxylin and eosin from a 90-day repeated oral dose toxicity study of Phy-Blica-D extract are displayed.** Panels A, E, I, and M show normal histology of kidney, liver, spleen, and lung tissue, respectively, in control rats at the end of the study period. Panels B, F, J, and N depict tissue sections from rats treated with 5 mg/kg of Phy-Blica-D extract; C, G, K, and O with 50 mg/kg; and D, H, L, and P with 300 mg/kg.

hygroscopicity, a characteristic previously associated with gum Arabic due to its high affinity for moisture and its molecular structure rich in hydrophilic polysaccharides. This property causes microcapsules containing higher GA proportions to absorb more atmospheric water during storage, potentially compromising their stability and shelf life. For example, Ahmadi et al. (2025) demonstrated that clove extract microcapsules containing higher ratios of GA showed significantly greater moisture uptake after 7 days compared to those encapsulated with maltodextrin alone [32]. Similarly, Akhavan Mahdavi et al. (2016) reported that gum Arabic increased water sorption and reduced flowability in anthocyanin powders [33]. These findings are consistent with our results, where microcapsules with a 6:4 MD:GA ratio (e.g., CE180_6:4) showed the highest hygroscopicity (15.51% after 7 days), reinforcing the need to optimize wall material composition to balance bioactive preservation and physical stability. Color characteristics were essential determinants of consumer acceptance, with higher MD ratios yielding lighter, aesthetically preferable powders. Conversely, higher GA proportions resulted in darker and yellower powders, negatively affecting consumer perceptions. These findings align with prior studies that emphasized

consumer preference for lighter-colored herbal beverages [34,35]. Therefore, the use of maltodextrin alone or combined minimally with GA is advisable to ensure desirable visual attributes.

The antioxidant activity assessments identified the microcapsule formulation CE180_6:4 as exhibiting superior performance across multiple assays (DPPH, ABTS, FRAP), suggesting effective preservation of phenolic antioxidants. The optimized GA proportions significantly contributed to the preservation of these antioxidants, aligning with prior research on the protective effects of GA against the thermal degradation of bioactive compounds [26,30,36]. Interestingly, crude extracts displayed consistently higher antioxidant potency than phenolic-rich fractions, as expected due to the lower content of core-coating material in the microcapsule powder of the phenolic-rich extracts compared to the microcapsule powder of the PBO decoction.

Although CE140_6:4 achieved the highest TPC (102.05 mg GA/g extract), this did not correspond to the strongest antioxidant activity. In contrast, CE180_6:4 consistently showed better antioxidant results in DPPH, ABTS, and FRAP tests, even with a slightly lower TPC. This indicates that antioxidant effectiveness relies not only on phenolic content but also on the stability and retention of bioactive compounds during spray drying. The higher inlet temperature of 180°C appears to help preserve the bioactivity of phenolics in the CE180_6:4 formulation.

Furthermore, this study also reveals sub-chronic toxicity information of a PBO-based tea blend, PBD. This repeated 90-day oral toxicity study is one of the studies employed in evaluating the safety of food ingredients, pharmaceuticals, and various chemical substances. In addition to the safety information obtained in this study, PBD is well-accepted by consumers, demonstrates significant *in vitro* antioxidant capacity, and shows no consistent treatment-related toxicological changes following repeated oral administrations over 28 days [9,10,15,16]. This should also be emphasized as a promising functional beverage.

Phy-Blica-D comprises a combination of various medicinal herbs, including *Phyllanthus emblica*, *Terminalia arjuna*, *Terminalia bellirica*, *Cyperus rotundus*, *Maerua siamensis*, *Tinospora crispa*, *Terminalia citrina*, *Allium sativum*, *Piper retrofractum*, *Zingiber officinale*, *Alpinia galangal*, *Glycyrrhiza glabra*, *Solanum torvum*, and *Aegle marmelos* [15,16]. These selected herbs have demonstrated notable antioxidant properties, functioning as both primary and secondary antioxidants. Several herbs, such as *P. emblica* [37], *T. arjuna* [38], *T. bellirica* [39], *T. crispa* [40], *A. sativum* [41], and *Z. officinale* [42] are widely supported by animal and clinical studies for their beneficial biological effects, including antioxidant, glucose-lowering, lipid-lowering, anti-inflammatory, and liver-protective activities. Additionally, herbs like *G. glabra* [43] and *A. marmelos* [44], chosen for their taste-enhancing attributes, also exhibit established antioxidant activities and confirmed safety profiles. Our previous study has revealed several bioactive compounds, including 6-galloylglucose, ferulic acid, naringenin, glycyrrhizic acid, and 6-gingerol, found in PBD [16]. These components have been reported *in vivo* and in clinical studies for their remarkable antioxidant properties and other health benefits [45–47]. Therefore, these compounds are proposed as suitable biological markers contributing to the overall antioxidant efficacy of Phy-Blica-D.

The findings from the ongoing sub-chronic toxicity assessment show that the No-Observed-Adverse-Effect Level (NOAEL) for the PBD decoction exceeds 300 mg/kg body weight per day, equating to roughly 3.7 L/kg body weight daily. No signs of organ-specific toxicity were detected in either male or female rats. The NOAEL of 300 mg/kg/day in rats corresponds to a human equivalent dose (HED) of 50.27 mg/kg/day based on standard conversion [48]. Given that the extract concentration in PBD was approximately 83.8 mg/100 mL (0.838 mg/mL), this HED equates to an approximate beverage intake of 600 mL/kg/day. As mentioned in our previous article, 12 of the 14 herbal components indicated $LD_{50}$ values between 1,000 and 5,000 mg/kg body weight [16]. The Globally Harmonized System of Classification and Labelling of Chemicals (GHS) classifies 10 of the 14 herbs in PBD—specifically *P. emblica*, *T. arjuna*, *T. bellirica*, *C. rotundus*, *T. crispa*, *T. citrina*, *A. sativum*, *Z. officinale*, *A. galanga*, and *S. torvum*—into category 5, signifying relatively low acute toxicity with $LD_{50}$ values over 2,000 mg/kg body weight and up to 5,000 mg/kg [16]. Therefore, the current results reinforce the safety of the PBD formulation, establishing an essential scientific basis for determining dosing parameters for

future *in vivo* studies, which will include assessments of its hypoglycemic, hypolipidemic, and antioxidant effects on non-communicable diseases.

PBO and PBD are recognized as traditional herbal tonics that warrant further investigation to emphasize their potential biological activity and safety as functional ingredients or beverage products. Although this study provides extensive evidence on the antioxidant potential, physicochemical properties, and sub-chronic toxicity profile of the developed polyherbal functional beverage, some limitations remain. The existing safety assessment primarily relied on animal models, specifically rats. Consequently, the findings, especially those related to toxicity and dosing advice, require validation through future clinical studies involving human participants. Furthermore, this study did not thoroughly evaluate the long-term stability of the encapsulated herbal powder under actual commercial storage conditions, which is crucial for establishing practical shelf-life guidelines. Significantly, the increased hygroscopicity seen in CE180_6:4 (15.51% after 7 days) poses a storage challenge. To address this, future formulations might lower the GA proportion or add encapsulation stabilizers. Additionally, enhancing commercial viability could involve moisture-barrier packaging (such as aluminum foil laminates with desiccants) and cold-chain storage systems.

## 5. Conclusions

This study successfully developed and optimized PBO into a functional ingredient using spray-drying microencapsulation. Optimal encapsulation efficiency was achieved at 140°C with maltodextrin as the wall material. At the same time, superior antioxidant activity was observed in formulations prepared at 180°C with a maltodextrin:gum Arabic ratio of 6:4. These encapsulated powders demonstrated favorable physicochemical properties, which may contribute to improving their stability and bioactivity. The resulting formulation, PBD, showed high consumer acceptability, potent in vitro antioxidant activity, and no observed adverse effects in a 90-day sub-chronic oral toxicity study, with a NOAEL exceeding 300 mg/kg/day. These results confirm the potential of PBD as a safe, antioxidant-rich functional beverage. For practical utilization, this research supports developing PBD as a commercially viable product within the functional food and complementary medicine industries. The encapsulated powder could be incorporated into ready-to-drink teas, dietary supplements, or wellness formulations targeting oxidative stress-related health concerns. However, further evaluation of the long-term storage stability of microencapsulated powders under real-world conditions, along with the bioavailability and pharmacokinetics of the key bioactive compounds *in vivo*, is necessary for manufacturing and application.

## Supporting information

**S1 Table. Composition of *Phyllanthus emblica*-based functional herbal tea with varying proportions of herbs and spices (grams of each plant per 2.5 g).**
(DOCX)

**S2 Table. Sensory data for functional herbal tea based on *Phyllanthus emblica* with varying proportions of herbs and spices.**
(DOCX)

**S1 Fig. The particle size distribution profiles of microencapsulated Phy-Blica-O powders, prepared by spray drying at various inlet temperatures (140°C: panels A–C; 180°C: panels D–F) and different maltodextrin (MD) to gum Arabic (GA) ratios: (A, D) MD:GA (10:0), (B, E) MD:GA (8:2), and (C, F) MD:GA (6:4) are presented.** Particle sizes are displayed as differential volume (%) and cumulative volume (%) distribution curves.
(DOCX)

**S2 Fig. The particle size distribution profiles of microencapsulated polyphenol-rich extracts from Phy-Blica-O powders were prepared by spray drying at various inlet temperatures (140°C: panels A–C; 180°C: panels D–F)**

and different maltodextrin (MD) to gum Arabic (GA) ratios: (A, D) MD:GA (10:0), (B, E) MD:GA (8:2), and (C, F) MD:GA (6:4). Particle sizes are displayed as differential volume (%) and cumulative volume (%) distribution curves.
(DOCX)

**S3 Fig. The antioxidant activity of Phy-Blica herbal tea formulations is assessed by the inhibition of fluorescence decay in fluorescein.** Fluorescence decay curves illustrate the relative fluorescence intensity (%) over 90 minutes in the presence of various sample concentrations (µg/mL): (A) Trolox (reference antioxidant), (B) Phy-Blica-O, (C) Phy-Blica-B, (D) Phy-Blica-D, and (E) Phy-Blica-E. A more significant inhibition of fluorescence decay indicates more potent antioxidant activity.
(DOCX)

## Author contributions

**Conceptualization:** Thammarat Kaewmanee, Acharaporn Issuriya, Sasitorn Chusri.

**Data curation:** Thammarat Kaewmanee, Acharaporn Issuriya, Piyapong Choochana, Pinanong Na-Phatthalung, Sasitorn Chusri.

**Formal analysis:** Piyapong Choochana, Surasak Limsuwan.

**Funding acquisition:** Thammarat Kaewmanee, Sasitorn Chusri.

**Methodology:** Acharaporn Issuriya, Piyapong Choochana.

**Project administration:** Sasitorn Chusri.

**Supervision:** Surasak Limsuwan, Sasitorn Chusri.

**Visualization:** Pinanong Na-Phatthalung.

**Writing – original draft:** Thammarat Kaewmanee, Acharaporn Issuriya, Piyapong Choochana, Pinanong Na-Phatthalung, Surasak Limsuwan, Sasitorn Chusri.

**Writing – review & editing:** Thammarat Kaewmanee, Acharaporn Issuriya, Pinanong Na-Phatthalung, Surasak Limsuwan, Sasitorn Chusri.

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
