## [Decision Letter · Decision Letter 0]

30 May 2025

Dear Dr. Chusri,

Thank you for submitting your manuscript to PLOS ONE. After careful consideration, we feel that it has merit but does not fully meet PLOS ONE’s publication criteria as it currently stands. Therefore, we invite you to submit a revised version of the manuscript that addresses the points raised during the review process.

We look forward to receiving your revised manuscript.

Kind regards,

Shengqian Sun

Academic Editor

PLOS ONE

Journal Requirements:

This research was supported by the Prince of Songkla University (Pattani campus) (Grant No. SAT601363S) and was partially supported by Mae Fah Luang University (Grant No. 651C5003). Mae Fah Luang University covered the article processing fees charge.

This research was supported by the Prince of Songkla University (Pattani campus) (Grant No. SAT601363S) and was partially supported by Mae Fah Luang University (Grant No. 651C5003). Mae Fah Luang University covered the article processing fees charge.

This research was supported by the Prince of Songkla University (Pattani campus) (Grant No. SAT601363S) and was partially supported by Mae Fah Luang University (Grant No. 651C5003). Mae Fah Luang University covered the article processing fees charge.

7. Your ethics statement should only appear in the Methods section of your manuscript. If your ethics statement is written in any section besides the Methods, please delete it from any other section.

Reviewers' comments:

Reviewer's Responses to Questions

**Comments to the Author**

1. Is the manuscript technically sound, and do the data support the conclusions?

Reviewer #1: Yes

Reviewer #2: Yes

2. Has the statistical analysis been performed appropriately and rigorously?

Reviewer #1: Yes

Reviewer #2: Yes

3. Have the authors made all data underlying the findings in their manuscript fully available?

Reviewer #1: Yes

Reviewer #2: Yes

4. Is the manuscript presented in an intelligible fashion and written in standard English?

Reviewer #1: Yes

Reviewer #2: Yes

Reviewer #1: PLOS ONE

Functional beverage development from traditional polyherbal tonic: Antioxidant-rich microcapsules and comprehensive sub-chronic toxicity assessment

General comment: The manuscript in reference compiles the reported “Functional beverage development from traditional polyherbal tonic: Antioxidant-rich microcapsules and comprehensive sub-chronic toxicity assessment”. The manuscript has relevant information and organization that will be interesting for readers. However, some points need to be addressed before further consideration.

1. Abstract: The writing style should be concise, clear, and reflect the objectives, research methods, results, conclusions, and suggestions in full.

2. Introduction:

-What are the highlights of this article? Should add details.

3. Method:

-Should be shown a flow chart as an illustration of 2.1, 2.2, and 2.3.

-Should be added details of equipment in this study, including model, brand, and country of manufacture.

4. Discussion:

- “Optimal encapsulation efficiencies for the PBO and its phenolic-rich extracts were achieved at lower drying temperatures (140°C), with higher proportions of MD, consistent with previous research highlighting the effectiveness of maltodextrin in preserving bioactive components during drying processes [26-28].” Should be explained the role of maltodextrin in preserving bioactive components during the drying process.

-“Nevertheless, formulations with increased GA content exhibited elevated

hygroscopicity, a characteristic previously associated with GA, underscoring the necessity of selecting appropriate encapsulation materials [34, 35].” It should be clearly explained, including examples of related research and its relationship to the results obtained in this research.

5. Conclusion:

-Should be rewritten and summarized important points that consistent with the article objectives

-Should be added suggestions, utilization, and guidelines for future development.

6. Other Suggestions:

-Should be checked the reference format according to the journal's requirements.

-Should be updated references.

Reviewer #2: The manuscript presents a comprehensive study on the development of a functional beverage, Phy-Blica-D (PBD), derived from a traditional Thai polyherbal tonic, Phy-Blica-O (PBO), through microencapsulation and sub-chronic toxicity assessment.However, the manuscript has some limitations, including incomplete statistical reporting, limited discussion of encapsulation trade-offs, and minor inconsistencies in terminology and data presentation. With revisions, the manuscript could make a significant contribution to the field of functional beverages and complementary medicine:

1. Title:

Revise the short title to correct the grammatical error: “Phy-Blica-O and Phy-Blica-D: Development of Microcapsules and Sub-Chronic Toxicity Assessment.”

Consider specifying “Thai” in the main title for clarity: “Functional Beverage Development from Traditional Thai Polyherbal Tonic...”

Ensure keyword consistency (e.g., capitalize “Sub-Chronic Toxicity” to match others).

2. Abstract:

Clarify the antioxidant activity claim by specifying which assays showed superiority for CE180_6:4 or acknowledge variability across assays.

Correct or explain the HED calculation to ensure unit consistency (e.g., convert to volume based on beverage concentration).

Specify the organs examined in the toxicity study for transparency.

3. Introduction:

Streamline the functional food discussion to focus on herbal beverages and their specific benefits.

Justify the choice of spray-drying by briefly comparing it to other encapsulation methods or citing its advantages (e.g., scalability, cost-effectiveness).

Clarify the cytotoxicity findings (e.g., specify concentrations tested) to strengthen the safety claim.

There is no explanation why encapsulation is important for PBO and PBD.

There is no explanation why GA and MD were used for encapsulation of PBO and PBD.

Please explain in details the gap in the research before the objectives.

4. Materials and methods:

Justify the solvent choices and pH adjustments for phenolic extraction, citing relevant literature or preliminary experiments.

Explain the selection of inlet temperatures (140°C, 180°C) based on prior studies or thermal stability of PBO components.

What about the single treatment of MD or GA not in a combination.

Clarify the statistical analysis for sensory data (e.g., ANOVA, significance level) and report results in a table or supplementary file.

Include weekly body weight and food consumption trends in the toxicity study (e.g., as a supplementary figure) to show temporal patterns.

5. Results and discussion:

Provide detailed statistical results (e.g., p-values, effect sizes) for antioxidant and toxicity data, and discuss significant comparisons in the text.

Acknowledge the trade-off between TPC (higher in CE140_6:4) and other antioxidant assays (higher in CE180_6:4), and explain why CE180_6:4 was deemed optimal.

Discuss potential solutions for high hygroscopicity (e.g., modified packaging, lower GA ratios) to enhance commercial viability.

Include a table or figure summarizing sensory scores and panelist preferences to support claims about PBD’s consumer acceptability.

Clarify the HED calculation, specifying the beverage concentration (e.g., mg/mL) and correcting the volume-based expression (mL/kg).

6. Conclusions:

Balance the claim about CE180_6:4 by noting the TPC advantage of CE140_6:4 and clarifying the basis for optimality.

Correct the HED expression and provide the beverage concentration used for the volume estimate.

Expand the limitations to include stability testing and scalability challenges, with suggestions for future research.

**Do you want your identity to be public for this peer review?** For information about this choice, including consent withdrawal, please see our Privacy Policy

Reviewer #1: No

Reviewer #2: **Yes: ** Saeid Jafari

---

## [Author Response · Author response to Decision Letter 1]

6 Aug 2025

Reviewer #1: PLOS ONE

Functional beverage development from traditional polyherbal tonic: Antioxidant-rich microcapsules and comprehensive sub-chronic toxicity assessment”

General comment: The manuscript in reference compiles the reported “Functional beverage development from traditional polyherbal tonic: Antioxidant-rich microcapsules and comprehensive sub-chronic toxicity assessment”. The manuscript has relevant information and organization that will be interesting for readers. However, some points need to be addressed before further consideration.

1. Abstract: The writing style should be concise, clear, and reflect the objectives, research methods, results, conclusions, and suggestions in full.

Author Response: We sincerely thank the reviewer for their valuable suggestion. In response, we have revised the abstract to clearly and concisely present the study’s objective, methodology, key findings, conclusion, and potential applications in the revised version.

2. Introduction: -What are the highlights of this article? Should add details.

Author Response: Thank you for this insightful suggestion. We have revised the Introduction section to clearly highlight the main points and importance of the study. Specifically, we have (1) clarified the reason for transforming Phy-Blica-O (a traditional polyherbal tonic) into a functional ingredient through microencapsulation, (2) emphasized the novelty of combining optimization of spray-drying conditions with extensive physicochemical and antioxidant assessments, (3) highlighted the scientific and regulatory importance of conducting a 90-day sub-chronic oral toxicity study to ensure safety for human use, and (4) stressed the translational potential of PBD as a safe, antioxidant-rich functional beverage suitable for use in complementary and integrative medicine.

3. Method:

-Should be shown a flow chart as an illustration of 2.1, 2.2, and 2.3.

-Should be added details of equipment in this study, including model, brand, and country of manufacture.

Author Response: We appreciate the reviewer for the helpful suggestion. In response, we have added a flow chart, now labeled as Figure 1, to visually illustrate the experimental workflow described in Sections 2.1 (Preparation of Phy-Blica-O herbal tea), 2.2 (Development of decoction and phenolic-rich extracts), and 2.3 (Microencapsulation via spray-drying). This illustration is intended to enhance reader comprehension of the stepwise procedures and clarify the overall methodology.

Furthermore, we have thoroughly revised the Methods section to include the full specifications of all major equipment used in the study. This consists of the model number, brand name, and country of manufacture for equipment involved in freeze drying, spray-drying, particle analysis, microscopy, colorimetry, moisture, and water activity measurements.

4. Discussion:

- “Optimal encapsulation efficiencies for the PBO and its phenolic-rich extracts were achieved at lower drying temperatures (140°C), with higher proportions of MD, consistent with previous research highlighting the effectiveness of maltodextrin in preserving bioactive components during drying processes [26-28].” Should be explained the role of maltodextrin in preserving bioactive components during the drying process.

Author Response: We thank the reviewer for this helpful suggestion. In response, we have expanded the relevant sentence in the Discussion section to clarify the mechanistic role of maltodextrin (MD) in preserving bioactive compounds during spray-drying as follows.

“Optimal encapsulation efficiencies for the PBO and its phenolic-rich extracts were achieved at lower drying temperatures (140°C), with higher proportions of MD, consistent with previous research highlighting the effectiveness of maltodextrin in preserving bioactive components during drying processes [26-28]. This protective role is attributed to its low viscosity, high water solubility, and excellent film-forming ability, which contribute to forming a stable encapsulating matrix around heat-sensitive phytochemicals, thereby minimizing thermal degradation and oxidation during spray-drying [26–28].”

-“Nevertheless, formulations with increased GA content exhibited elevated

hygroscopicity, a characteristic previously associated with GA, underscoring the necessity of selecting appropriate encapsulation materials [34, 35].” It should be clearly explained, including examples of related research and its relationship to the results obtained in this research.

Author Response: We appreciate the reviewer’s insightful suggestion. In response, we have revised the discussion to explain the hygroscopic properties of gum Arabic better, supported by literature and linked to our findings as follows.

“Nevertheless, formulations with increased GA content exhibited elevated hygroscopicity, a characteristic previously associated with gum Arabic due to its high affinity for moisture and its molecular structure rich in hydrophilic polysaccharides. This property causes microcapsules containing higher GA proportions to absorb more atmospheric water during storage, potentially compromising their stability and shelf life. For example, Ahmadi et al. (2025) demonstrated that clove extract microcapsules containing higher ratios of GA showed significantly greater moisture uptake after 7 days compared to those encapsulated with maltodextrin alone [34]. Similarly, Akhavan Mahdavi et al. (2016) reported that gum Arabic increased water sorption and reduced flowability in anthocyanin powders [35]. These findings are consistent with our results, where microcapsules with a 6:4 MD:GA ratio (e.g., CE180_6:4) showed the highest hygroscopicity (15.51% after 7 days), reinforcing the need to optimize wall material composition to balance bioactive preservation and physical stability.”

5. Conclusion:

-Should be rewritten and summarized important points that consistent with the article objectives

-Should be added suggestions, utilization, and guidelines for future development.

Author Response: We thank the reviewer for this valuable recommendation. In response, we have revised the Conclusion section to better reflect the core objectives of the study, which are (i) optimizing microencapsulation of Phy-Blica-O (PBO) and (ii) evaluating the antioxidant capacity and sub-chronic toxicity of Phy-Blica-D (PBD), as well as incorporating recommendations for future research, potential applications, and product development guidelines as follows.

“This study successfully developed and optimized PBO into a functional ingredient using spray-drying microencapsulation. Optimal encapsulation efficiency was achieved at 140°C with maltodextrin as the wall material. At the same time, superior antioxidant activity was observed in formulations prepared at 180°C with a maltodextrin:gum Arabic ratio of 6:4. These encapsulated powders demonstrated favorable physicochemical properties, which may contribute to improve their stability and bioactivity. The resulting formulation, PBD, showed high consumer acceptability, potent in vitro antioxidant activity, and no observed adverse effects in a 90-day sub-chronic oral toxicity study, with a NOAEL exceeding 300 mg/kg/day. These results confirm the potential of PBD as a safe, antioxidant-rich functional beverage. For practical utilization, this research supports developing PBD as a commercially viable product within the functional food and complementary medicine industries. The encapsulated powder could be incorporated into ready-to-drink teas, dietary supplements, or wellness formulations targeting oxidative stress-related health concerns. However, further evaluation of the long-term storage stability of microencapsulated powders under real-world conditions, along with the bioavailability and pharmacokinetics of the key bioactive compounds in vivo, is necessary for manufacturing and application.”

6. Other Suggestions:

-The reference should be checked according to the journal's requirements.

-Should be updated references.

Author Response: We sincerely thank the reviewer for this important comment. In response:

- Reference Format: All references have been carefully reviewed and reformatted to fully comply with PLOS ONE's citation style, which follows the Vancouver style with the following structure: Author(s). Title. Journal Abbreviation. Year;Volume(Issue):Page range. doi (if available).

- Updated References: We have updated and replaced outdated or superseded citations with more recent publications (mostly from the past 5 years). This ensures the manuscript reflects the current state of research in functional food development, microencapsulation, and polyherbal safety assessment.

- DOI Inclusion: Where applicable, DOIs have been added to all journal articles, in accordance with PLOS ONE’s submission guidelines. We believe these adjustments improve both the academic rigor and alignment with the journal’s formatting standards.

Reviewer #2: The manuscript presents a comprehensive study on the development of a functional beverage, Phy-Blica-D (PBD), derived from a traditional Thai polyherbal tonic, Phy-Blica-O (PBO), through microencapsulation and sub-chronic toxicity assessment. However, the manuscript has some limitations, including incomplete statistical reporting, limited discussion of encapsulation trade-offs, and minor inconsistencies in terminology and data presentation. With revisions, the manuscript could make a significant contribution to the field of functional beverages and complementary medicine:

1. Title: Revise the short title to correct the grammatical error: “Phy-Blica-O and Phy-Blica-D: Development of Microcapsules and Sub-Chronic Toxicity Assessment.” Consider specifying “Thai” in the main title for clarity: “Functional Beverage Development from Traditional Thai Polyherbal Tonic...” Ensure keyword consistency (e.g., capitalize “Sub-Chronic Toxicity” to match others).

Author Response: We appreciate the reviewer’s detailed and helpful suggestions. For the Short Title, we have revised it to correct the grammatical error and improve clarity and conciseness. The original “Phy-Blica-O and Phy-Blica-D: Development of Microcapsules and Sub-Chronic Toxicity Assessment” has been updated to "Development of Microcapsules and Sub-Chronic Toxicity of Phy-Blica Formulations."

Furthermore, we agree with the suggestion to clarify the geographic and traditional context by specifying "Thai" in the main title. Therefore, the main title was changed from ‘Functional Beverage Development from Traditional Polyherbal Tonic: Antioxidant-Rich Microcapsules and Comprehensive Sub-Chronic Toxicity Assessment’ to ‘Functional Beverage Development from Traditional Thai Polyherbal Tonic: Antioxidant-Rich Microcapsules and Comprehensive Sub-Chronic Toxicity Assessment’

We have ensured consistency by capitalizing "Sub-Chronic Toxicity" to align with the formatting of other keywords. We believe these revisions enhance the clarity, specificity, and consistency of the manuscript presentation and are now fully aligned with journal expectations.

2. Abstract: Clarify the antioxidant activity claim by specifying which assays showed superiority for CE180_6:4 or acknowledge variability across assays. Correct or explain the HED calculation to ensure unit consistency (e.g., convert to volume based on beverage concentration). Specify the organs examined in the toxicity study for transparency.

Author Response: We sincerely thank the reviewer for this thoughtful and constructive feedback. In response, we have revised the Abstract to address all points raised:

- Antioxidant Activity of CE180_6:4: We have clarified that the CE180_6:4 microcapsule formulation showed superior antioxidant activity based on specific in vitro assays. The revised sentence now states:“Maximum encapsulation efficiency was achieved at 140°C with MD-only wall materials, while microcapsules with a 6:4 MD:GA ratio at 180°C exhibited the highest antioxidant activity based on DPPH, ABTS, and FRAP assays.” This provides clear attribution of assay-specific performance rather than implying universal superiority.

- Human Equivalent Dose (HED): To ensure unit consistency, we have clarified how the HED was derived from the NOAEL and converted to an approximate beverage volume based on the concentration used in the study. The revised sentence reads: “The calculated human equivalent dose is 50.27 mg/kg/day, which corresponds to an estimated volume of approximately 600 mL/kg/day based on the extraction yield obtained from the beverage.”

- Organs Examined in Toxicity Study: We have added detail on the organs examined for histopathological analysis to improve transparency. The sentence now reads: “The oral no-observed-adverse-effect level (NOAEL) exceeded 300 mg/kg/day, with normal histopathological profiles in the liver, kidneys, and spleen.”

3. Introduction: Streamline the functional food discussion to focus on herbal beverages and their specific benefits. Justify the choice of spray-drying by briefly comparing it to other encapsulation methods or citing its advantages (e.g., scalability, cost-effectiveness). Clarify the cytotoxicity findings (e.g., specify concentrations tested) to strengthen the safety claim. There is no explanation why encapsulation is important for PBO and PBD. There is no explanation why GA and MD were used for encapsulation of PBO and PBD. Please explain in details the gap in the research before the objectives.

Author Response: We thank the reviewer for these insightful comments. In response, we have carefully revised the Introduction section to address each point and strengthen the scientific rationale of the study. The following amendments have been made:

- Streamlining the functional food discussion: We revised the beginning of the introduction to specifically highlight herbal beverages as a subgroup of functional foods. The updated paragraph emphasizes their contribution to health, antioxidant properties, and traditional uses, rather than broadly discussing all types of functional foods.

- Justification for spray-drying: We have now included a brief comparison of encapsulation methods and provided the rationale for using spray-drying, with relevant citations added to support this justification.

- Clarification of cytotoxicity findings: We have expanded the discussion of our previous cytotoxicity work by stating ‘Among these formulas, Phyllanthus emblica-based functional herbal tea (THP-R016 or Phy-Blica-O (PBO)) possessed notable antioxidant properties and did not exhibit cytotoxic effects on Vero cells at concentrations up to 100 µg/mL, supporting its safety for oral applications [12, 13].’

- Importance of encapsulation for PBO and PBD: We have added a paragraph to explain why encapsulation is critical, stating that: ‘The utilization of microencapsulation in PBO not only enhances the sensory profile but also protects heat-sensitive bioactive compounds from degradation, thereby improving the functional properties and shelf-life of the formulation [14].’

- Rationale for using gum Arabic (GA) and maltodextrin (MD): We have now provided a detailed rationale, as follows. ‘Additionally, maltodextrin (MD) and gum Arabic (GA) are common wall materials in spray-drying botanical extracts because of their excellent film-forming, emulsifying, and protective qualities [15-17]. MD helps achieve high encapsulation efficiency and maintains low viscosity, whereas GA improves stability and retains antioxidants. Using them together has been demonstrated to increase encapsulation yields and functional performance in earlier research [15–17].’

4. Materials and methods: Justify the solvent choices and pH adjustments for phenolic extraction, citing relevant literature or preliminary experiments. Explain the selection of inlet temperatures (140°C, 180°C) based on prior studies or thermal stability of PBO components. What about the single treatment of MD or GA not in a combination. Clarify the statistical analysis for sensory data (e.g., ANOVA, significance level) and report results in a table or supplementary file. Include weekly body weight and food consumption trends in the toxicity study (e.g., as a supplementary figure) to show temporal patterns.

Author Response: We appreciate the reviewer’s detailed and

---

## [Decision Letter · Decision Letter 1]

26 Aug 2025

Dear Dr. Chusri,

Thank you for submitting your manuscript to PLOS ONE. After careful consideration, we feel that it has merit but does not fully meet PLOS ONE’s publication criteria as it currently stands. Therefore, we invite you to submit a revised version of the manuscript that addresses the points raised during the review process.

We look forward to receiving your revised manuscript.

Kind regards,

Shengqian Sun

Academic Editor

PLOS ONE

Journal Requirements:

Reviewers' comments:

Reviewer's Responses to Questions

**Comments to the Author**

Reviewer #1: All comments have been addressed

Reviewer #2: (No Response)

2. Is the manuscript technically sound, and do the data support the conclusions?

Reviewer #1: Yes

Reviewer #2: Yes

3. Has the statistical analysis been performed appropriately and rigorously?

Reviewer #1: Yes

Reviewer #2: Yes

4. Have the authors made all data underlying the findings in their manuscript fully available?

Reviewer #1: Yes

Reviewer #2: Yes

5. Is the manuscript presented in an intelligible fashion and written in standard English?

Reviewer #1: Yes

Reviewer #2: Yes

Reviewer #1: This article has been edited based on feedback.

This article is suitable for publication in PLOS ONE.

Reviewer #2: Major Comments

Statistical Reporting – While p-values and comparisons are included, effect sizes and confidence intervals should be more explicitly presented to strengthen result interpretation.

Encapsulation Trade-offs – The discussion should better highlight the balance between high TPC (CE140_6:4) and superior antioxidant activity (CE180_6:4), clarifying the rationale for selecting the “optimal” formulation.

Hygroscopicity Limitation – The practical challenges of high GA content (storage instability) should be more explicitly connected to commercial implications and possible solutions.

Toxicity Study Reporting – Weekly body weight and food intake trends are added, but a brief summary of variability and overall trajectory in the main text would enhance clarity.

Minor Comments

Abstract: Slightly long—could be streamlined for clarity, focusing on objective, key findings, and implications.

Introduction: Nicely revised, though still somewhat broad. Consider reducing general functional food market data and focusing more directly on herbal beverages.

Figures/Tables: Supplementary figures and tables are well presented. Ensure that all abbreviations are defined at first mention.

References: Updated references are appropriate, but a few older citations remain; consider substituting with more recent (past 5 years) where possible.

Language: Overall clear, but some sentences could be shortened to improve readability

**Do you want your identity to be public for this peer review?** For information about this choice, including consent withdrawal, please see our Privacy Policy

Reviewer #1: No

Reviewer #2: **Yes: ** Saeid Jafari

---

## [Author Response · Author response to Decision Letter 2]

30 Nov 2025

Dear Editors,

On behalf of my co-authors, I am pleased to resubmit our revised manuscript titled: “Functional Beverage Development from Traditional Thai Polyherbal Tonic: Antioxidant-Rich Microcapsules and Comprehensive Sub-Chronic Toxicity Assessment” (Manuscript ID: PONE-D-25-18265R2) for reconsideration for publication in PLOS ONE. We would like to express our sincere gratitude to the editorial team and reviewers for their thoughtful and constructive feedback. We have thoroughly addressed all comments and suggestions in the revised manuscript and the accompanying point-by-point response document, as listed at the end of the cover letter.

We believe that the revised version significantly improves the clarity, methodological transparency, and scientific rigor of our work, aligning with PLOS ONE’s editorial standards and scope. I appreciate your consideration and look forward to your favorable response.

Best Regards,

Sasitorn Chusri, Ph.D.

Corresponding Author

Biomedical Technology Research Group for Vulnerable Populations,

School of Health Science, Mae Fah Luang University

Muang, Chiang Rai 57100, Thailand

E-mail: sasitorn.chu@mfu.ac.th

Reviewer #2: Major Comments

1. Statistical Reporting – While p-values and comparisons are included, effect sizes and confidence intervals should be more explicitly presented to strengthen result interpretation.

Address to the reviewer’s comment: We thank you for this helpful suggestion. In response, we've revised the Results section to explicitly report effect sizes with 95% confidence intervals where relevant. These updates help readers understand both the magnitude and precision of the effects, going beyond just p-values. Additionally, we've updated tables to show effect sizes alongside statistical results.

2. Encapsulation Trade-offs – The discussion should better highlight the balance between high TPC (CE140_6:4) and superior antioxidant activity (CE180_6:4), clarifying the rationale for selecting the “optimal” formulation.

Address to the reviewer’s comment: Thank you for this insightful comment. We appreciate the reviewer’s suggestion to more clearly explain the trade-offs between total phenolic content (TPC) and antioxidant activity and to clarify the rationale behind selecting the optimal formulation.

In response, we have revised the Results and Discussion to explain how these factors influenced the formulation choice explicitly. Specifically, we now note that although CE140_6:4 showed the highest TPC (102.05 mg GA/g extract), this did not necessarily indicate the best antioxidant performance. Instead, microcapsules produced at 180°C with a 6:4 MD:GA ratio (CE180_6:4) consistently exhibited stronger antioxidant activity in DPPH, ABTS, and FRAP assays, indicating better retention and stability of key bioactive compounds at higher drying temperatures. This suggests that antioxidant strength depends not only on the amount of phenolics but also on their chemical stability after encapsulation. This information was added to the results and discussion sections.

3. Hygroscopicity Limitation – The practical challenges of high GA content (storage instability) should be more explicitly connected to commercial implications and possible solutions.

Address to the reviewer’s comment: Thank you for your insightful comment. We agree that the practical impact of high hygroscopicity in GA-rich formulations should be more clearly linked to commercial considerations and potential mitigation strategies. Therefore, we expanded the discussion to emphasize how high GA levels, especially in formulations like CE180_6:4, which had the highest hygroscopicity (15.51% after 7 days), present real challenges for product stability during storage, handling, and distribution. The updated text now underscores that excessive moisture uptake can lead to particle caking, decreased flowability, reduced antioxidant effectiveness, and shorter shelf life, which are key factors for the commercial application of microencapsulated herbal ingredients. These modifications enhance the Discussion section by clearly connecting scientific results to practical formulation challenges and industry solutions.

To address these challenges, we included clearer solutions such as

(1) decreasing GA content or adjusting MD: GA ratios to balance phenolic protection and physical stability

(2) incorporating stabilizing co-encapsulants

(3) using moisture-barrier packaging methods (e.g., aluminum foil laminates, desiccants, low-permeability films).

4. Toxicity Study Reporting – Weekly body weight and food intake trends are added, but a brief summary of variability and overall trajectory in the main text would enhance clarity.

Address to the reviewer’s comment: Thank you for your helpful comment. We agree that summarizing the weekly body weight and food-intake trends in the main text would enhance clarity and readability. Therefore, we have expanded the toxicity Results section to include a brief narrative that emphasizes the overall variability and trajectory observed over the 90-day study. The revised text now indicates that both male and female rats in all treatment groups experienced consistent, gradual increases in body weight over 90 days, without any abnormal fluctuations or deviations from the control group. Weekly food intake patterns remained stable and similar across all groups, with only minor weekly variations typical of normal physiological cycles. Notably, no dose-dependent effects were observed in either parameter, reinforcing the absence of treatment-related toxicity. This summary improves the clarity of the weekly data while remaining aligned with the detailed figures provided.

Minor Comments

5. Abstract: Slightly long—could be streamlined for clarity, focusing on objective, key findings, and implications.

Address to the reviewer’s comment: Thank you for this constructive comment. We appreciate the suggestion to make the abstract clearer and more focused. In response, we have revised the abstract to be more concise, highlighting the study’s main goal, the key findings from microencapsulation optimization and toxicity testing, and the main implications for developing functional beverages.

6. Introduction: Nicely revised, though still somewhat broad. Consider reducing general functional food market data and focusing more directly on herbal beverages.

Address to the reviewer’s comment: Thank you for the insightful comment. We agree that the Introduction section would benefit from a tighter focus on herbal beverages rather than a broader discussion of the functional food market. We removed several sentences and citations related to overall trends in the functional food market. This revision offers a clearer context for the study by directly linking PBO and its microencapsulation to current research and development in herbal-based functional drinks.

7. Figures/Tables: Supplementary figures and tables are well presented. Ensure that all abbreviations are defined at first mention.

Address to the reviewer’s comment: Thank you for your positive feedback on the supplementary figures and tables. We appreciate the reviewer’s reminder regarding the clarity of abbreviations. In response, we reviewed the entire manuscript, including the supplementary figures, tables, and captions, to ensure that all abbreviations are defined at first use, covering methodological terms, assays, sample codes, and parameters. These updates improve clarity and consistency.

8. References: Updated references are appropriate, but a few older citations remain; consider substituting with more recent (past 5 years) where possible.

Address to the reviewer’s comment: Thank you for highlighting the importance of the updated references. We reviewed the reference list and replaced many older citations with recent studies (from the past 5 years), especially in areas like functional herbal beverages, microencapsulation, and antioxidant assessment. We retained a few older references that are still seminal or foundational for traditional Thai polyherbal formulations and microencapsulation methodology, where no recent alternatives exist. This blend of updated and classic references offers a current yet historically grounded context for the work.

9. Language: Overall clear, but some sentences could be shortened to improve readability.

Address to the reviewer’s comment: Thank you for this helpful comment. We have carefully re-read the entire manuscript, focusing on sentence length and readability. In response, we shortened several long, multi‑clause sentences, especially in the Introduction, Discussion, and Methods sections, and removed minor redundancies to improve flow without changing the scientific meaning.

---

## [Editor Report · Decision Letter 2]

9 Dec 2025

Functional beverage development from traditional Thai polyherbal tonic: Antioxidant-rich microcapsules and comprehensive sub-chronic toxicity assessment

PONE-D-25-18265R2

Dear Dr. Chusri,

We’re pleased to inform you that your manuscript has been judged scientifically suitable for publication and will be formally accepted for publication once it meets all outstanding technical requirements.

Kind regards,

Shengqian Sun

Academic Editor

PLOS One

Additional Editor Comments (optional):

Although the manuscript is suitable for acceptance, one of the reviewers has still provided some comments, and we hope you could further polish the manuscript according to these suggestions to enhance its quality.

Reviewers' comments:

Statistical Reporting – While p-values and comparisons are included, effect sizes and confidence intervals should be more explicitly presented to strengthen result interpretation.

Encapsulation Trade-offs – The discussion should better highlight the balance between high TPC (CE140_6:4) and superior antioxidant activity (CE180_6:4), clarifying the rationale for selecting the “optimal” formulation.

Hygroscopicity Limitation – The practical challenges of high GA content (storage instability) should be more explicitly connected to commercial implications and possible solutions.

Toxicity Study Reporting – Weekly body weight and food intake trends are added, but a brief summary of variability and overall trajectory in the main text would enhance clarity.

Minor Comments

Abstract: Slightly long—could be streamlined for clarity, focusing on objective, key findings, and implications.

Introduction: Nicely revised, though still somewhat broad. Consider reducing general functional food market data and focusing more directly on herbal beverages.

Figures/Tables: Supplementary figures and tables are well presented. Ensure that all abbreviations are defined at first mention.

References: Updated references are appropriate, but a few older citations remain; consider substituting with more recent (past 5 years) where possible.

Language: Overall clear, but some sentences could be shortened to improve readability

---

## [Editor Report · Acceptance letter]

PONE-D-25-18265R2

PLOS One

Dear Dr. Chusri,

I'm pleased to inform you that your manuscript has been deemed suitable for publication in PLOS One. Congratulations! Your manuscript is now being handed over to our production team.

Kind regards,

on behalf of

Dr. Shengqian Sun

Academic Editor

PLOS One